# Thermochemical anomalies in the upper mantle control Gakkel Ridge accretion

John M. O'Connor [1,2,3 ✉], Wilfried Jokat [2,4], Peter J. Michael [5], Mechita C. Schmidt-Aursch [2], Daniel P. Miggins[6] & Anthony A. P. Koppers [6]

Despite progress in understanding seafloor accretion at ultraslow spreading ridges, the ultimate driving force is still unknown. Here we use $^{40}Ar/^{39}Ar$ isotopic dating of mid-ocean ridge basalts recovered at variable distances from the axis of the Gakkel Ridge to provide new constraints on the spatial and temporal distribution of volcanic eruptions at various sections of an ultraslow spreading ridge. Our age data show that magmatic-dominated sections of the Gakkel Ridge spread at a steady rate of ~11.1 ± 0.9 mm/yr whereas amagmatic sections have a more widely distributed melt supply yielding ambiguous spreading rate information. These variations in spreading rate and crustal accretion correlate with locations of hotter thermo-chemical anomalies in the asthenosphere beneath the ridge. We conclude therefore that seafloor generation in ultra-slow spreading centres broadly reflects the distribution of ther-mochemical anomalies in the upper mantle.

[1] GeoZentrum Nordbayern, University Erlangen-Nürnberg, Schlossgarten 5, 91054 Erlangen, Germany. [2] Alfred Wegener Institute Helmholtz Centre for Polar and Marine Research, Am Handelshafen 12, 27570 Bremerhaven, Germany. [3] Faculty of Science, Vrije Universiteit Amsterdam, De Boelelaan 1085, 1081 HV Amsterdam, Netherlands. [4] University of Bremen, Fachbereich 5 - Geosciences, 28359 Bremen, Germany. [5] College of Engineering & Natural Sciences, University of Tulsa, Tulsa, OK 74104, USA. [6] College of Earth, Ocean, and Atmospheric Sciences, Oregon State University, Corvallis, OR 97331-5503, USA.
✉email: j.m.oconnor@vu.nl

Solving questions about the relation between the Earth's lithospheric plates and the asthenosphere is crucial for a more advanced understanding of mantle convection processes and how they might be directly, or indirectly, related to plate tectonics. Although the Gakkel Ridge extends for 1800 km under the Arctic sea ice it is nevertheless an ideal place to address these questions because it contains no significant transform offsets that offer a barrier to mantle flow[1]. In 2001 a high-resolution mapping and rock sampling study of Gakkel Ridge was accomplished during the international AMORE ice-breaker cruise to the high Arctic and North Pole[1]. Before this expedition, it was predicted—based on extrapolations from slow- to fast-spreading ridges—that ultraslow spreading would produce sparse volcanism, thinner crust, and very little hydrothermal activity, and low extents of melting of the underlying mantle[1].

However, the findings from the AMORE cruise contradicted some of these predictions by showing that hydrothermal activity

is far more abundant and that the ridge can be divided into two 'magmatism dominated' western (WVZ) and eastern (EVZ) zones separated by a 'sparsely magmatic zone' (SMZ)[1,2] (Fig. 1). Moreover, the oceanic crust is highly variable with thick crust below the magmatic transverse ridges and thin crust in between in the more amagmatic zones[2,3]. Crucially, if the spreading rate decreases systematically along the Gakkel Ridge from ~13 mm/yr in the west to only ~6 mm/yr in the east at 120 °E[4–6], then the prediction of a global relationship between spreading rate and crustal thickness is no longer valid at ultraslow spreading rates. In this context, the formation of separate individual magmatic and amagmatic segments at ultraslow ridges is an important discovery, yet their origin cannot be explained by (local) differences in the rate of seafloor spreading and plate motion[1,2,7].

Thus far, the lack of precise age dating on young (<5 Ma) oceanic crust prevents any detailed understanding of such processes along mid-ocean ridges. Our study is a first attempt to shed

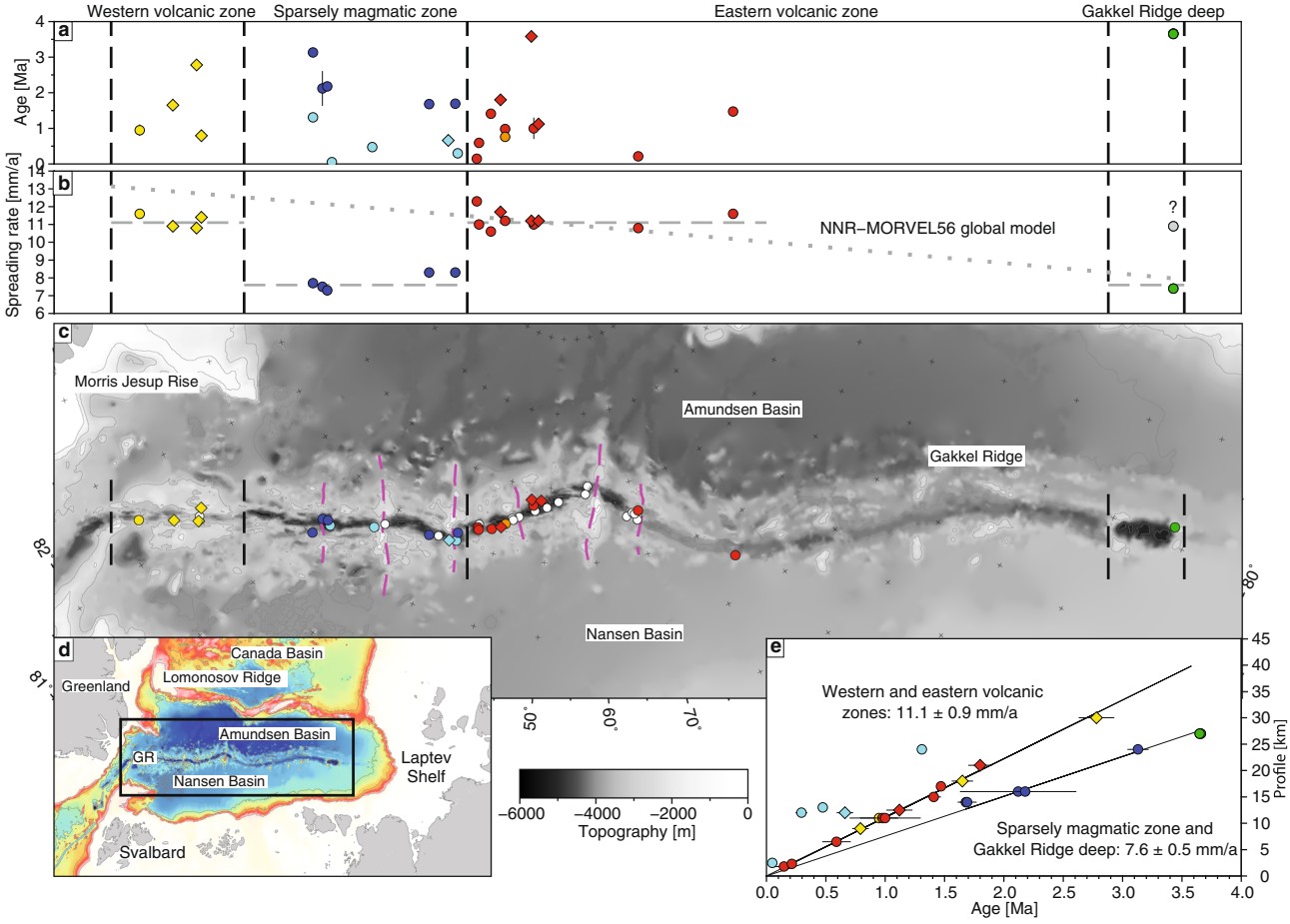

**Fig. 1 Relation between spreading rate and magmatism along the Gakkel Ridge.** All panels refer to Table 1 and Figs. 2–5. Yellow, blue, red and green symbols represent samples with reliable spreading rates. Light blue symbols are for samples that do not yield realistic spreading rates due to their mode of emplacement. The orange dot marks a sample without a profile. The diamond symbols are for samples with dates that fail on the data quality criteria discussed in the 'Methods' section. Error bars for age data are 2σ uncertainty. Black dashed lines mark the boundaries of the different volcanic regimes. **a** Distribution of isotopic ages along Gakkel Ridge **b** Distribution of calculated spreading rates along the ridge. The question mark and open symbol are for an inferred 'fast' spreading rate at the eastern end of the GRD. The 'slow' spreading rate estimated for the eastern end of the GRD corresponds to the abrupt narrowing of the rift valley extending into the Laptev Sea. Thus, the 'slow' spreading rate calculated for the eastern end of the GRD likely continues to the east. The dotted line is for average spreading rates for the past 3 Ma calculated with the NNR-MORVEL56 global model[8]. **c** Topography of the Gakkel Ridge[56] and the distribution of dated samples along the ridge. Smaller white dots are for samples that did not yield ages. Pink dashed lines are for well-defined centres of focussed magmatism[10]. **d** Map of the study area, GR; Gakkel Ridge. **e** Average spreading rates calculated using the combined age-distance profiles. A robust linear regression (forced through the origin) yields an average spreading rate of 11.1 ± 0.9 mm/yr for the WMZ and EVZ. A poorly constrained rate of 7.6 ± 0.5 mm/yr is inferred for four SMZ and a single GRD-LS sample whereas the other SMZ samples do not predict a consistent spreading rate. An average of all the SMZ data, excluding the sample that does not meet all the data quality criteria, also yields a slow rate (8.1 mm/yr). Note that in some cases samples with large errors overlap high-precision data. This figure was created with GMT[57].

more light on these questions by reporting on a systematic $^{40}$Ar/$^{39}$Ar dating study along the ultraslow spreading Gakkel Ridge. In doing so, we aim to test the fundamental assumption based on aeromagnetic data that spreading rates decrease systematically from ~13 to ~6 mm/yr along this mid-ocean ridge. This is essential because thus far only aeromagnetic data have been collected along the Gakkel Ridge due to ice coverage. Flight lines that are typically acquired at several hundred metres altitude and spaced around 18 km apart cannot resolve small magnetic anomalies, particularly those related to very young ages[6]. Arctic kinematic age models based on aeromagnetic data, therefore, are interpolated, tending to smooth out differences in spreading rate and type, and they do not provide reliable identification of anomalies younger than Chron 5 (<10 Ma). Barring any presence of these artefacts, modern global models of seafloor spreading rates such as NNR-MORVEL56[8] provide the best estimate of spreading rates along the Gakkel Ridge (Fig. 1).

Here, we present evidence that other drivers—such as the thermal and chemical structure of the underlying mantle—may control the development of the two endmember-type magmatic and amagmatic spreading segments[1,3,7,9] and that advances our understanding of the processes controlling ultraslow spreading on a global scale.

## Results

**$^{40}$Ar/$^{39}$Ar isotopic dating**. Here we report the first extensive $^{40}$Ar/$^{39}$Ar dataset that can robustly measure spreading rates along spreading centres, even along an ultraslow spreading ridge. To this end, we have drawn on recent advances in high-resolution multi-collector mass spectrometry to measure $^{40}$Ar/$^{39}$Ar isotopic dates for young low-potassium basalt samples dredged from the Gakkel Ridge during the AMORE cruises in 2001 (Fig. 1). The results of twenty-five $^{40}$Ar/$^{39}$Ar incremental heating dates for acid-leached basaltic groundmass are summarised in Table 1 and Supplementary Table 1 and Supplementary Dataset 1. Full analytical results are available in Supplementary Dataset 2. The criteria for assessing the quality of the age data are discussed in the 'Methods' section. Seven samples do not meet one or both of the criteria that a plateau consists of at least 50% $^{39}$Ar and a $P$-value >5% (Table 1 and Fig. 1). These seven samples therefore cannot be considered robust but rather approximate age estimates.

**Spreading rate calculation**. We have used the isotopic dates to define a range of age profiles across the Gakkel Ridge rift valley between the present day and 3.7 Ma (Fig. 1 and Table 1). Individual age-distance profiles across the ridge and calculated spreading rates are shown in Figs. 2–5. The various sources of uncertainty in dredge-sampling an ultraslow spreading ridge are considered in the 'Methods' section. We can minimise this dredging uncertainty by stacking multiple age-distance profiles when calculating average spreading rates for the different segments of the ridge (Fig. 1). We argue that, notwithstanding the various assumptions and sources of uncertainty ('Methods' section), the $^{40}$Ar/$^{39}$Ar dates provide high-precision constraint on spreading rates along the magmatic WVZ and EVZ sections of the ridge: 11 ± 0.9 mm/yr ($n = 15$) (Fig. 1). In contrast, the $^{40}$Ar/$^{39}$Ar dates for the SMZ and Gakkel Ridge Deep (GRD)-Laptev Sea (LS)[10] group at faster and slower spreading rates, respectively, rather than falling along a single spreading rate (Fig. 1). Four SMZ and a single Gakkel Ridge Deep (GRD) sample predict a poorly constrained slower spreading rate: 7.6 ± 0.5 mm/yr ($n = 7$) (Fig. 1). The other SMZ dated samples predicting un-realistically fast-spreading rates were dredged from near the base of the rift valley wall (Fig. 3) suggesting there is significant off-axis magmatism (half the dated samples). This age distribution is

consistent with a wider zone of accretion rather than a slower spreading rate. The 'apparent' spreading rate for samples erupted on-axis would be slower because it will take longer for lavas erupted on-axis to move off-axis if spreading is accommodated over a wider zone. The same could be true for the GRD, which is based on a single dated sample. However, regressing all the SMZ age data, except for one sample that does not meet all data quality criteria (Table 1) also yields a slower spreading rate (8.1 mm/yr) compared to the WVZ and EVZ so it cannot be excluded that these dates might belong to two different groups. Moreover, none of the dates lies in the vicinity of the WVZ/EVZ regression, even when considering the error bars. Thus, while we cannot link the slower SMZ-GRD trend necessarily to plate separation, it may well prove to be significant for understanding amagmatic crustal accretion.

**Variation in spreading rate**. In the case of the Gakkel Ridge, we are observing a consistent spreading rate in the magmatic WVZ and EVZ segments (Fig. 1). A lack of transform faults along the Gakkel Ridge is well-documented[1] and is in accord with steady ~11 mm/yr spreading rates along most of the ridge (Fig. 1). Nevertheless, if we define the spreading rate as the plate separation rate, then it cannot vary (in terms of degrees of rotation about a pole) along the ridge as this would violate the geometric tenets of rigid plate tectonics. Therefore, differing spreading rates —such as the poorly constrained ~7.6 mm/yr rate we infer for the SMZ—can only be a young and unstable process. Otherwise, we would expect to observe large offsets between segments. This raises the possibility of another form of tectonic dislocation and/ or differences in the magmatic evolution in amagmatic segments.

For example, the SMZ dates might reflect a different style of lithospheric/crustal accretion. This notion is consistent with evidence from axial volcanic ridge morphology and abyssal hill patterns suggesting this cyclic process is the normal mode of development of axial parts of slow[11,12] and ultraslow ridges[13]. In the WVZ, this process is evident from the transport of mid-Atlantic type elongate volcanic ridges away from the rift zone, e.g., refs. [1,2] (Fig. 2). In the EVZ, larger and slightly more magmatic edifices are dismembered, with basalts dominating the seafloor in between[1,2] (Fig. 4). As a result, when calculating the WVZ and EVZ spreading rates we are using dated samples that have been dredged from magmatic structures that have been fragmented apart and transported away as the seafloor spreads from the active central rift axis. Because of this style of discontinuous volcanism, it is possible to measure the spreading rate based on the rifting of discreet volcanoes and ridges that have been erupted in the rift valley (see 'Methods' section for specific examples).

In contrast, seafloor spreading in the SMZ involves very little axial volcanism and is likely to be mostly amagmatic, with fresh mantle peridotites emplaced directly on the seafloor at the spreading axis[1] (Fig. 3). Five of the 10 dates reported for the SMZ give anomalously young ages for their location in the rift valley (Fig. 3). Samples PS59 311-2 (Profile 10; 661 ± 68 ka) and PS59 310-001-1 (295 ± 53 ka) are located near the base of the rift valley wall where the rift valley bisects a transverse volcanic basement ridge[1,2,10] (Fig. 3). Sample PS59/252-1 (Profile 8; 474 ± 30 ka) is located near the rift valley wall on the southern flank of a large volcanic centre that is less rifted—compared to the well-developed rift valley to the north and south—and also seems to be related to a transverse volcanic basement ridge (Fig. 3). Sample HLY0102-D36-1 (Profile 7; 49 ± 34 ka) is located on a volcanic structure that straddles the active rift axis implying that it is a recent eruption (Fig. 3). Samples HLY0102-D35-12 (Profile 6; 1.31 ± 0.03 Ma) and sample HLY0102-D35-20 (3.13 ± 0.09 Ma)

**Table 1 Summary of $^{40}$Ar/$^{39}$Ar age data and calculated full spreading rates.**

| Profile | Station | Sample | Long | Lat | Age Ma/ka | Error 2σ | Age profiles km | Off-axis distance km | Spreading Rate mm/yr | K₂O wt% | Age Ma | Error 2σ | ³⁹Ar % | P-value | MSWD |
|---|---|---|---|---|---|---|---|---|---|---|---|---|---|---|---|
| *WVZ* | | | | | | | | | | | | | | | |
| Profile 1 | PS59/223 | -27 | -4.3983 | 83.3000 | 948 | ±160 ka | 11.0 | 9.1 | 11.6 | 0.236 | 0.95 | 0.16 | 100 | 98 | 0.51 |
| Profile 2 | ***PS59/226*** | -23 | -2.1983 | 83.6967 | 1.65 | ±0.09 Ma | 18.0 | 6.3 | 10.9 | 0.251 | 1.65 | 0.09 | **45** | 5 | 1.85 |
| Profile 3 | ***PS59/231*** | 20 | -0.4267 | 83.9650 | 792 | ±66 ka | 9.0 | 6.5 | 11.4 | 0.357 | 0.79 | 0.07 | 88 | **1** | 1.83 |
| Profile 4 | ***HLY0102/D24*** | -5 | -1.6560 | 84.0918 | 2.78 | ±0.15 Ma | 30.0 | 14.1 | 10.8 | 0.240 | 2.78 | 0.15 | 91 | 4 | 1.66 |
| *SMZ* | | | | | | | | | | | | | | | |
| Profile 5 | PS59/244 | -2 | 11.047 | 85.030 | 2.12 | ±0.49 Ma | 16.0 | 13.9 | 7.5 | 0.304 | 2.12 | 0.49 | 100 | 100 | 0.16 |
| Profile 6 | HLY0102/D35 | -20 | 10.7347 | 85.2571 | 3.13 | ±0.09 Ma | 24.0 | 11.6 | 7.7 | 0.530 | 3.13 | 0.09 | 79 | 63 | 0.8 |
| Profile 6 | HLY0102/D35 | -12 | 10.7347 | 85.2571 | 1.31 | ±0.03 Ma | 24.0 | 11.6 | Fault related? | 0.530 | 1.31 | 0.03 | 68 | 56 | 0.89 |
| Profile 7 | HLY0102/D37 | -8 | 11.4920 | 85.2925 | 2.18 | ±0.09 Ma | 16.0 | 9.7 | 7.3 | 0.509 | 2.18 | 0.09 | 91 | 20 | 1.31 |
| Profile 7 | HLY0102/D36 | -1 | 12.3317 | 85.2587 | 49 | ±34 ka | 2.5 | 1.9 | Near rift axis | 0.750 | 0.05 | 0.03 | 99 | 84 | 0.71 |
| Profile 8 | PS59/252-1 | -1 | 18.2067 | 85.6200 | 474 | ±30 ka | 13.0 | 8.0 | Fault related? | 0.650 | 0.47 | 0.03 | 83 | 72 | 0.81 |
| Profile 9 | PS59/312 | -11 | 27.3383 | 85.9233 | 1.68 | ±0.05 Ma | 14.0 | 7.7 | 8.3 | 0.301 | 1.68 | 0.05 | 76 | 6 | 1.69 |
| Profile 10 | ***PS59/311*** | -2 | 31.2167 | 85.9767 | 661 | ±68 ka | 12.0 | 4.6 | Fault related? | 0.335 | 0.66 | 0.07 | 96 | **0** | 2.4 |
| No profile | PS59/310 | -001-1 | 32.51 | 86.009 | 295 | ±53 ka | 12.0 | 5.1 | Fault related? | 0.466 | 0.30 | 0.05 | 100 | 72 | 0.81 |
| Profile 11 | HLY0102/D95 | -38 | 32.219 | 86.116 | 1.69 | ±0.08 Ma | 14.0 | 6.1 | 8.3 | 0.481 | 1.69 | 0.08 | 79 | 55 | 0.9 |
| *EVZ* | | | | | | | | | | | | | | | |
| Profile 12 | HLY0102/D90 | -18 | 35.597 | 86.279 | 147 | ±82 ka | 1.8 | 0.9 | 12.0 | 0.449 | 0.15 | 0.08 | 100 | 100 | 0.29 |
| Profile 12 | PS59/309 | -39 | 35.9000 | 86.2500 | 590 | ±116 ka | 6.5 | 5.6 | 11.0 | 0.447 | 0.59 | 0.12 | 70 | 59 | 0.86 |
| Profile 13 | HLY0102/D89 | -4 | 38.398 | 86.315 | 1.41 | ±0.06 Ma | 15.0 | 7.1 | 10.6 | 0.429 | 1.41 | 0.06 | 61 | 41 | 1.03 |
| Profile 14 | ***HLY0102/D85*** | -90 | 40.169 | 86.372 | 1.80 | ±0.10 Ma | 21.0 | 6.5 | 11.7 | 0.407 | 1.80 | 0.1 | 92 | **0** | 2.63 |
| Profile 15 | PS59/305 | -1 | 40.9667 | 86.4283 | 978 | ±42 ka | 11.0 | 3.5 | 11.2 | 0.483 | 0.98 | 0.04 | 71 | 82 | 0.66 |
| No profile | PS59/305 | -20 | 40.9667 | 86.4283 | 758 | ±29 ka | 11.0 | | a | | 0.76 | 0.03 | 97 | 40 | 1.05 |
| Profile 16 | ***PS59/300*** | -18 | 45.6667 | 86.8167 | 3.58 | ±0.07 Ma | 40.0 | 16.0 | 11.2 | Rb=3.4 | 3.58 | 0.07 | **49** | 35 | 1.11 |
| Profile 16 | PS59/299 | -1 | 46.2700 | 86.7417 | 1.00 | ±0.30 Ma | 11.0 | 8.0 | 11.0 | 0.054 | 1.00 | 0.3 | 73 | 54 | 0.91 |
| Profile 17 | ***PS59/271-1*** | -14 | 47.942 | 86.818 | 1.12 | ±0.11 Ma | 12.5 | 9.6 | 11.2 | 0.426 | 1.12 | 0.11 | 88 | **4** | 1.73 |
| Profile 18 | PS59/276-1 | -5 | 70.3650 | 86.5950 | 213 | ±26 ka | 2.3 | 8.1ᵇ | 11.0 | 0.484 | 0.21 | 0.03 | 80 | 24 | 1.22 |
| Profile 19 | HLY0102/D66 | -32 | 84.5783 | 85.5353 | 1.47 | ±0.01 Ma | 17.0 | 12.0 | 11.6 | 0.845 | 1.47 | 0.01 | 95 | 45 | 1.0 |
| *GRD* | | | | | | | | | | | | | | | |
| Profile 20 | PS72-472 | -1 (a) | 121.521 | 81.2025 | 3.66ᶜ | ±0.01 Ma | 27.0 | 13 | 7.4 | 0.69ᵃ | 3.66 | 0.01 | 68 | 40 | 1.04 |
| Profile 20 | PS72-472 | -1c | 121.521 | 81.2025 | 3.65ᶜ | ±0.01 Ma | 27.0 | 13 | 7.4 | 0.69ᵃ | 3.65 | 0.01 | 100 | 54 | 0.94 |
| | D23 | -1 | -0.8650 | 84.0100 | No age | | | 1.4 | | 0.227 | | | | | |
| | DS1 | -29 | 43.4852 | 86.5575 | No age | | | 2.0 | | 0.220 | | | | | |
| | DS2 | -2 | 42.1172 | 86.5224 | No age | | | 3.8 | | 0.303 | | | | | |
| | DS5 | -9 | 59.3847 | 87.0292 | No age | | | 1.9 | | 0.075 | | | | | |
| | DS7 | -2 | 67.5790 | 86.5518 | No age | | | 3.5 | | 0.254 | | | | | |
| | DS8 | -10 | 69.9288 | 86.5242 | No age | | | 0.4 | | 0.294 | | | | | |
| | D83 | -1 | 42.3597 | 86.5010 | No age | | | 1.3 | | 0.424 | | | | | |
| | D91 | -21 | 35.635 | 86.323 | No age | | | 2.2 | | 0.51 | | | | | |
| | PS59/254 | -1 | 19.4450 | 85.7383 | No age | | | | | Rb=6.7 | | | | | |
| | PS59/260 | -1 | 29.055 | 85.970 | No age | | | 4.2 | | 0.46 | | | | | |
| | PS59/270 | -36 | 49.512 | 86.735 | No age | | | 3.7 | | 0.439 | | | | | |
| | PS59/275-1 | -1 | 69.0450 | 86.5850 | No age | | | 4.6 | | 0.439 | | | | | |
| | PS59/289 | -69 | 69.7283 | 86.4767 | No age | | | 6.0 | | 0.260 | | | | | |
| | PS59/290-1 | -44 | 69.4200 | 86.5617 | No age | | | 2.8 | | 0.190 | | | | | |
| | PS59/293 | -17 | 58.6800 | 86.9233 | No age | | | 3.4 | | 0.300 | | | | | |
| | PS59/295 | -1 | 52.123 | 86.822 | No age | | | 3.0 | | 0.450 | | | | | |
| | PS59/297 | -37 | 46.8600 | 86.6750 | No age | | | 0.5 | | 0.156 | | | | | |

Bold italic indicate failed data quality criteria discussed in the 'Methods' section.
Italic indicate samples that were analysed but did not yield a 40Ar/39Ar date.
ᵃSlightly older age for PS59/305-1 versus PS59/305-20 used to calculate profile 15 (see Fig. 5).
ᵇRift valley filled with a large volcano.
ᶜData from ref. 10.

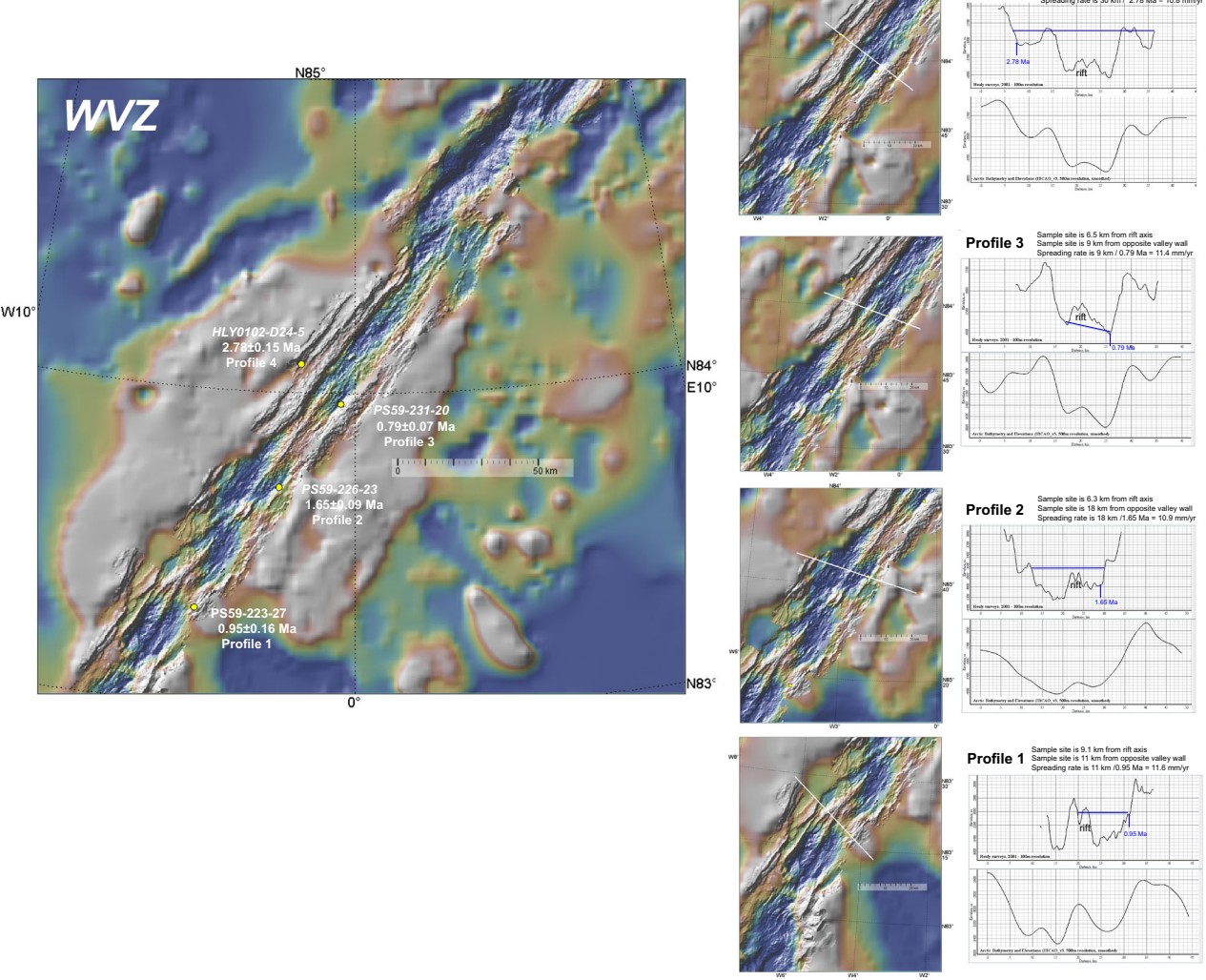

**Fig. 2 Age-distance profiles in the WVZ.** A large panel shows the locations of dredge samples and reported isotopic ages and an assigned age-distance profile number. The italicised sample labels are for dates that fail on the data quality criteria discussed in the 'Methods' section (Table 1). These individual profiles (P) are shown on smaller bathymetric maps as white lines crossing the rift valley. The locations of dated dredge samples are indicated by yellow dots. Profile depths are based on high-resolution multibeam (100-m contour) data (top panel) and the lower resolution International Bathymetric Chart of the Arctic Ocean (IBCAO)[56] (bottom panel). Shown also are the distances between dredge sites and the rift axis (labelled 'rift') and between the corresponding magmatic fragment on the opposite side of the rift, and the calculated full spreading rate (Table 1). Blue arrows are for the locations of dated samples along with depth profiles. P1 is orientated somewhat obliquely to the rift valley to connect the dating sample site with the corresponding fragment on the northern side of the rift. The fragment on the northern side is largely indistinguishable from the rift axis due to asymmetric spreading. The profile is orientated across this fragment where the axis and fragment can be distinguished at the same depth as the dated sample. P4 is slightly oblique to locate the flank of the opposite flank using available high-resolution multibeam data. P2 and P4 are located higher up in the flanks than the locations of the dated samples to take into account sloping rift valley walls. This location adjustment adds insignificant amounts to the length of the profiles and the calculated spreading rates (<1 km; <0.5 mm/yr). Map prepared in GeoMapApp (www.geomapapp.org).

are both from the same dredge haul located high up on the rift valley wall. All five young ages are evidence of a relatively low-volume transverse volcanic basement ridge extending away from the rift valley. Thus, the five anomalously young SMZ ages seem to be related to low-relief volcanism that can erupt across the entire rift valley and in association with transverse volcanic basement ridges (Fig. 3). Furthermore, low volume SMZ volcanism could also represent the across-axis "wandering" of axial magmatism over time.

## Discussion

In the magmatic WVZ and EVZ, much larger volcanic centres can fill the rift valley, and samples from their flanks are mostly

expected to yield ages consistent with rifting at the rift axis, e.g., ref. [1]. But if these large volcanic centres are sampled at their base closer to the rift valley wall, they could potentially yield anomalously young ages for their location in the rift valley and corresponding to a wider zone of accretion that is typically found in amagmatic segments. Large volcanic structures might also evolve differently after rifting relative to the location of the more amagmatic and normal rift valley walls. As discussed in the 'Methods' section a potential source of uncertainty in calculating the spreading rate is whether one or both walls facing the rift axis is sloping and leading to a variation in the calculated spreading rate. For example, the increased elevation of the magmatic WVZ and EVZ might result in valley walls with shallower profiles compared to the amagmatic SMZ. This will increase the wall

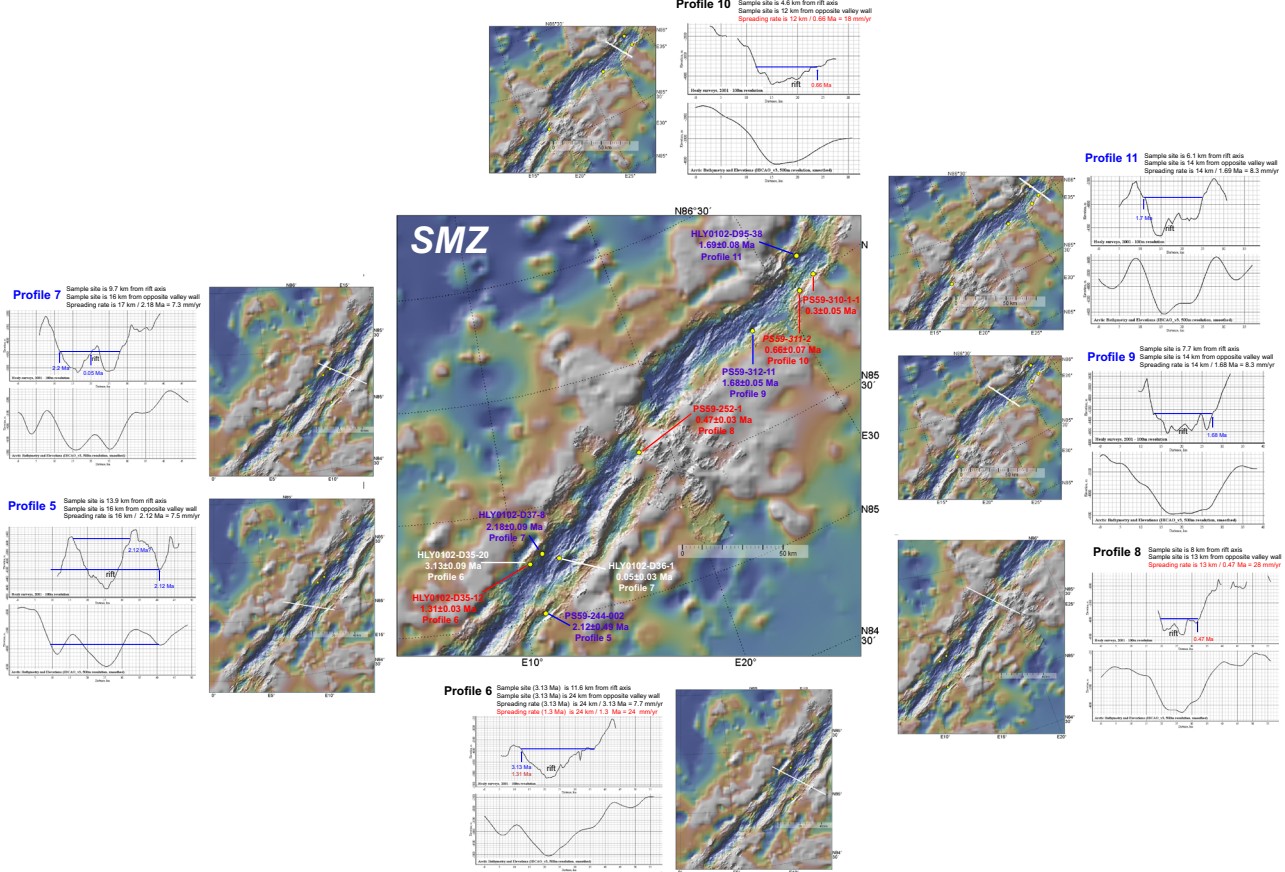

**Fig. 3 Age-distance profiles in the SMZ.** The SMZ is magma poor and magmatism can reach the surface away from the rift axis via large faults. P5 is oblique to the rift valley to connect the offset eastern tips of elongate fragments. A further complication is that the dredge sample was recovered on the side of a magmatic fragment facing away from the rift axis. Calculating the spreading rate from this location yields an anomalously fast-spreading rate of ~15 mm/yr compared to the rest of the profiles, particularly the nearby P6 and P7. The spreading rate is calculated from the crest of the fragment assuming that the dated sample reflects the age for the fragment as a whole. Red text is for samples from the valley walls yielding unrealistic young ages and spreading rates. For example, the dredge haul along P6 recovered basalts dated at 1.3 to 3.1 Ma. The anomalously young age is compatible with young basalts reflecting off-axis volcanism through faults. Blue text is for samples yielding coeval ages and matching spreading rates from opposite flanks at the southern (P5 & P7) and northern (P9 & P11) end of the SMZ. Other details as in Fig. 2.

separation for coeval rifted fragments leading to the calculation of a faster spreading rate. This mechanism might be expressed at the GRD by the inferred relationships between the spreading rate and the steepness of the valley walls (Fig. 5).

Alternatively, deep-reaching faults[14] and/or jumps of the rift axis suggested by asymmetric spreading (Table 1) might act to keep the SMZ rift valley wall more vertical. At ultraslow-spreading ridges, detachment faulting is thought to be a fundamental process in mantle rock exhumation and the generation of smooth seafloor[14–16]. In the case of the amagmatic SMZ, microearthquakes show that it lacks shallow seismicity in the upper 15 km of the lithosphere, but that it unusually produces earthquakes down to depths of 35 km[14]. This implies cold, thick lithosphere, with an upper aseismic zone that probably reflects substantial serpentinization of the upper lithosphere either along distinct, deep-reaching shear zones that concentrate strain or through pervasive alteration of at least 10% of the mantle rocks[14]. Evidence that in amagmatic regions of ultraslow-spreading ridges, serpentinization and fluid circulation may reach far deeper into the mantle than previously assumed is in line with less magmatism that isn't located so much along the central part of the axis but instead reflects a deeper and wider melting regime.

Next, we consider whether trends in our data correlate with published geophysical and geochemical data and attempt to understand the overall (regional) functioning of an ultraslow spreading ridge. More specifically, we are seeking a relationship between the new $^{40}Ar/^{39}Ar$ age data—especially the data that are inconsistent with spreading ages predicted by marine magnetic data—and the various unexpected findings arising from the AMORE expedition. A recent global shear wave velocity model[17] shows that temperature varies in the upper mantle beneath the Gakkel Ridge. The WVZ is located above a low-velocity anomaly (Fig. 6) reflecting the presence of hotter asthenosphere at a depth of 150–80 km in the Eurasia Basin[17] possibly due to mantle inflow from the North Atlantic[9,17,18]. According to this model, the EVZ is located above another seismic anomaly corresponding to hot asthenosphere at a depth of 200–150 km, which extends across the central Arctic from the Canada Basin to the Eurasia Basin (Fig. 6). The apparent agreement between the low-velocity anomalies and the boundaries of the magmatic and amagmatic segments is remarkable (Fig. 6). Moreover, they coincide better with an SMZ-EVZ boundary that we can infer from the differences in the average spreading rate and the style of crustal accretion, which is about 5°E east of initial estimates at 34°E[1] (Fig. 6). But at shallower asthenospheric depths the low-velocity anomalies do not match the magmatic and amagmatic boundaries (Supplementary Figure 2). It is generally the case, however, that the details of

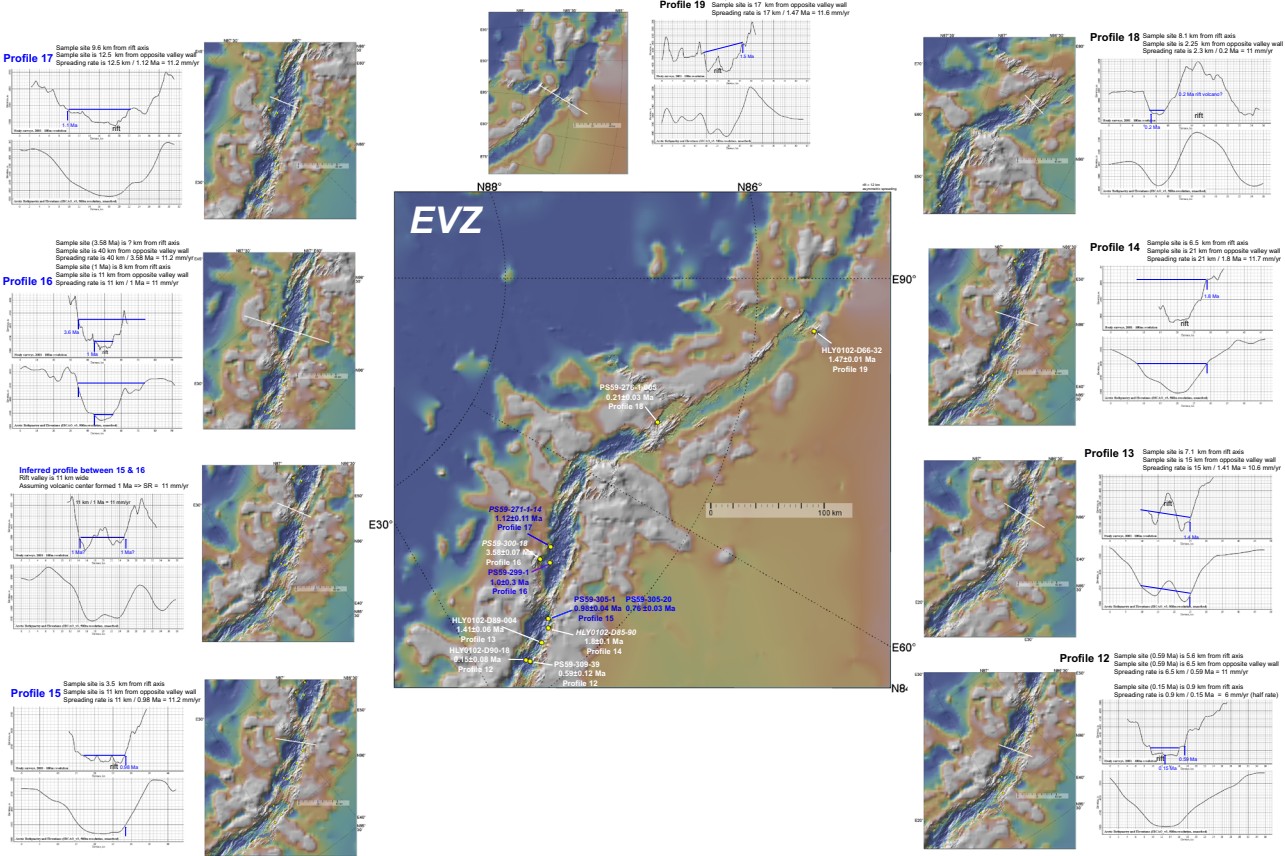

**Fig. 4 Age-distance profiles in the EVZ.** Blue text is for samples yielding coeval ~1 Ma ages and matching spreading rate from opposite flanks the southern (P15) and northern (P16 & P17) flanks of the EVZ. These profiles bound a magmatic centre in the rift valley. A similar spreading rate can be inferred from this volcanism assuming that P15–P17 samples are from the same event. The process of successive cycles of volcanism and rifting discussed in the text is illustrated by P18 marking the early stage of dismemberment of a 0.2 Ma large axial ridge-like volcano filling the rift valley and P19 crossing an older, already dismembered 1.5 Ma axial volcano. Other details as in Fig. 2.

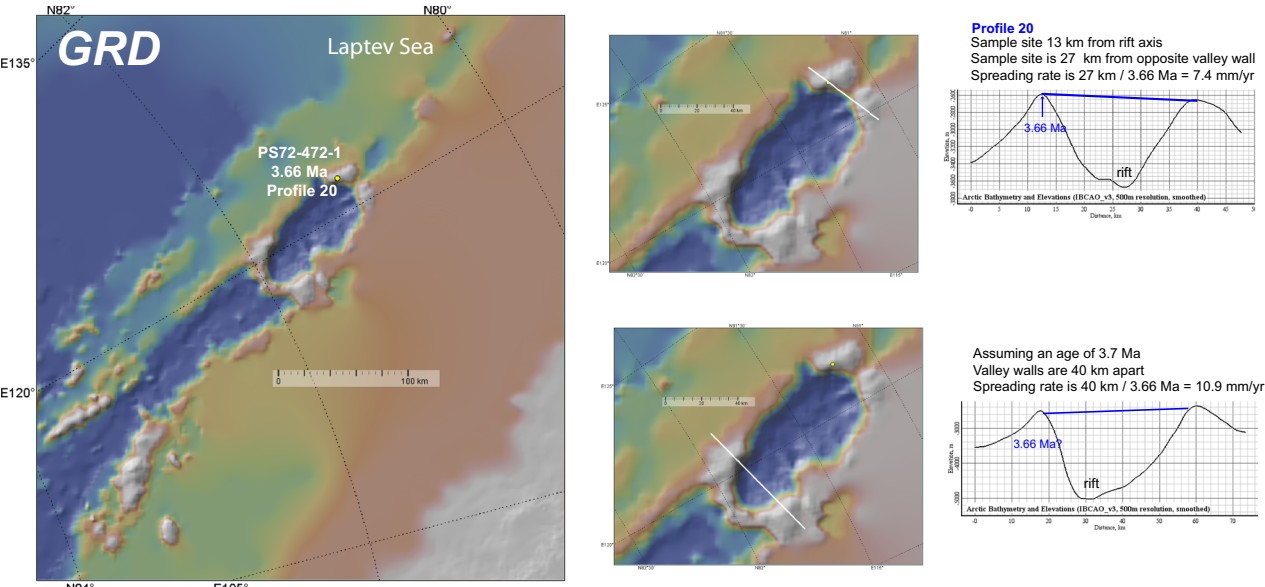

**Fig. 5 Age-distance profile in the GRD.** P20 predicts the dismemberment of a 3.7 Ma[10] axial volcano. Note that the wider rift valley to the west of P20 corresponds broadly with the width of the EVZ whereas the narrower rift valley east of P20 extends into the Laptev Sea. Furthermore, assuming that the volcanism on the flank of the wider axial valley at the western end of the GRD is also 3.7 Ma yields a faster EVZ-like spreading rate. Other details as in Fig. 2.

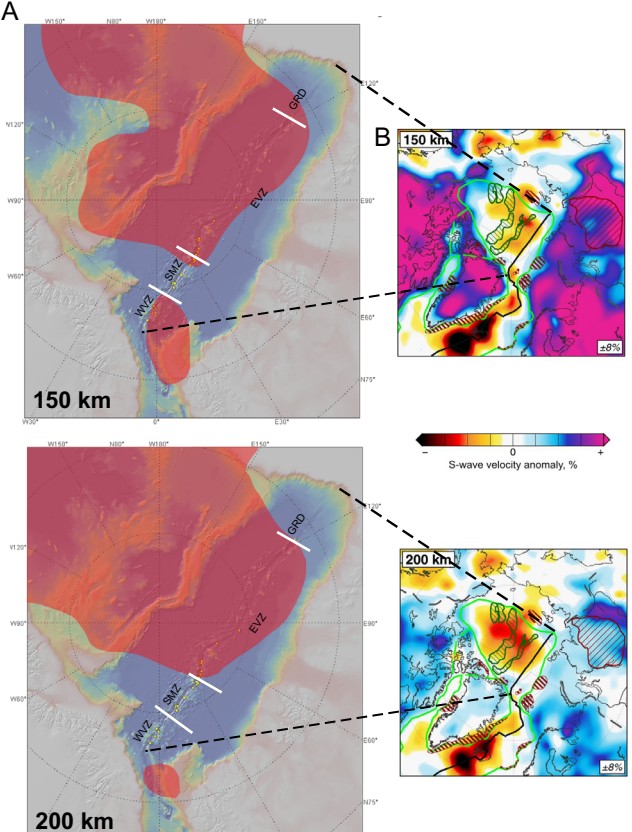

**Fig. 6 The Gakkel Ridge volcanic zones and upper mantle low-velocity anomalies. A** Relation between low-velocity anomalies at 200 and 150 km depth[17] and the different volcanic zones[1] as defined by average spreading rates in Fig. 1. Note that the low-velocity anomaly extending across the central Arctic underlies the EVZ and the low-velocity anomaly extending from the North Atlantic is located under the WVZ. The SMZ and GRD lie between these low-velocity regions. **B** The AMISvArc tomographic model at 200 and 150 km from ref. [17] and Supplementary Figure 2 show that these low-velocity anomalies do not extend to shallower depths. This tomographic model is a global, shear wave velocity model of the upper mantle and crust incorporating additional data from in and around the Arctic region[17]. Further details are available in ref. [17].

(deeper) structures are better constrained by surface wave tomography. For example, the wavelength of the data in the referenced study is around 230 km (Vs = 4.62 km/s, f = 50 s) so the tomography model can only resolve structures of half of the wavelength (~120 km). This is especially true in the central Arctic, where seismic stations are only located at the rim of the ocean basin. This low seismic station density limits even further the tomographic resolution and explains why the low-velocity anomalies do not match the WVZ, SMZ and EVZ boundaries at shallower mantle depths. Nevertheless, while there are potentially large errors in the tomography model for the shallow structure of the mantle under the ridge, it provides first-order information on the temperature below the Gakkel Ridge, which does correlate with our findings derived from the surface volcanism. We conclude therefore that there is a relation between a constant WVZ and EVZ spreading defined by $^{40}Ar/^{39}Ar$ dates and hotter regions in the mantle. We show also that this relationship breaks down above colder regions. Before this study, the assumption was that the spreading rate decreases steadily along the length of the ridge so ruling out the possibility of a relationship between spreading rate and mantle temperature.

A different study uses locally recorded microearthquakes to show that the lithosphere under amagmatic segments is much thicker (≤40 km) than below magmatic segments and only has a thin or absent oceanic crust[2,3,19]. Below the magmatic segments, the lithospheric thickness varies dramatically along-axis, thinning to as little as 15 km under volcanoes or sites of basalt exposure[14,19]. Thus, our $^{40}Ar/^{39}Ar$ dates reveal that spreading rate and accretion style correlates also with lithospheric thickness. Published geochemical data are also consistent with such a relationship. For example, a colder mantle leads to reduced melting as does a thicker lithosphere by creating a lid that inhibits asthenospheric upwelling, e.g., refs. [20–24]. In these situations, more fusible (enriched) components should be extracted preferentially with less dilution by liquids derived further up in the melting column[10] and references therein[25]. In the SMZ, low % melting shown by the lack of magmatic activity is also evident from the chemistry of the basalts[1] that exhibit a positive correlation between $Na_{8.0}$ ($Na_2O$ normalized to 8% MgO) and depth as a proxy for % melting[26,27]. This is contrasted by the shallower WVZ that exhibit the lowest $Na_{8.0}$ and the EVZ falling roughly in between the WVZ and SMZ[1]. The alkaline GRD sample (PS72/472-1) is predicted to lie above the common $Na_{8.0}$ range of the Gakkel Ridge (2.8–3.6 wt% $Na_2O$) pointing toward lower degrees of mantle melting[10]. Trace element and isotopic data for the ridge also point to a greater proportion of more fusible components due to low extents of melting at the SMZ[1] and GRD-LS segments[10], consistent with thicker lithosphere[14,19] and/or cooler mantle. In the amagmatic segments, any low-volume melts that get erupted may have been transported through the thickened lithosphere via faults, allowing for the transfer of uncontaminated low-volume melts of more fusible (fertile ± $CO_2$) mantle sources.

Radiogenic isotope ratios of Sr-Nd-Pb are not fractionated by magmatic processes and thus can help to more directly characterize the source(s) contributing to melts. Sr-Nd-Pb isotope ratios in WVZ lavas have a distinct isotopic signature (DUPAL[28]) considered to reflect contamination by thermally eroded subcontinental lithospheric mantle (SCLM)[18]. Contamination by SCLM and/or subducted lithosphere[29,30] might explain enhanced melting, e.g., ref. [31] that is consistent with the somewhat shallower low-velocity anomaly beneath the WVZ (Fig. 6). Thus, the correspondence between the northern limit of the observed SCLM-contaminated low-velocity anomaly and the observed boundary between the WVZ and SMZ suggests that the inflowing hot mantle may have modified the character of WVZ seafloor spreading[17,18]. In the case of the less magmatic EVZ (compared to the WVZ)[1], Sr-Nd-Pb isotope ratios and highly incompatible element ratios show that the mantle does not have a distinct (e.g., DUPAL) isotopic signature[10,32]. In contrast, it is suggested that on a local scale the Arctic mantle contains low-melting fertile components, e.g., refs. [10,18,20] and references therein[32]. Reference [32] modelled trace elements with mantle melting to find that geochemical variability in the EVZ can be explained by a heterogeneous mantle source composed of depleted MORB mantle plus a metasomatized mantle. Figure 7 shows that the maximum level of enrichment of this heterogenous EVZ mantle source increases systematically westward, roughly from the middle of the EVZ to the SMZ-WVZ boundary. In addition, we infer that the maximum level of source enrichment increases systematically eastwards toward the eastern boundary of the EVZ marked by the GRD (Fig. 7). We argue that this geochemical trend is more consistent with decreasing temperature than lithospheric thickness, particularly as geophysical studies show no evidence for a corresponding systematic thickening of the lithosphere[14,19].

Finally, ref. [18] identify a compositional boundary between the WVZ and EVZ mantle sources within the SMZ, at around 14°E (Fig. 7). The Sr-Nd-Pb isotopic composition of SMZ samples on

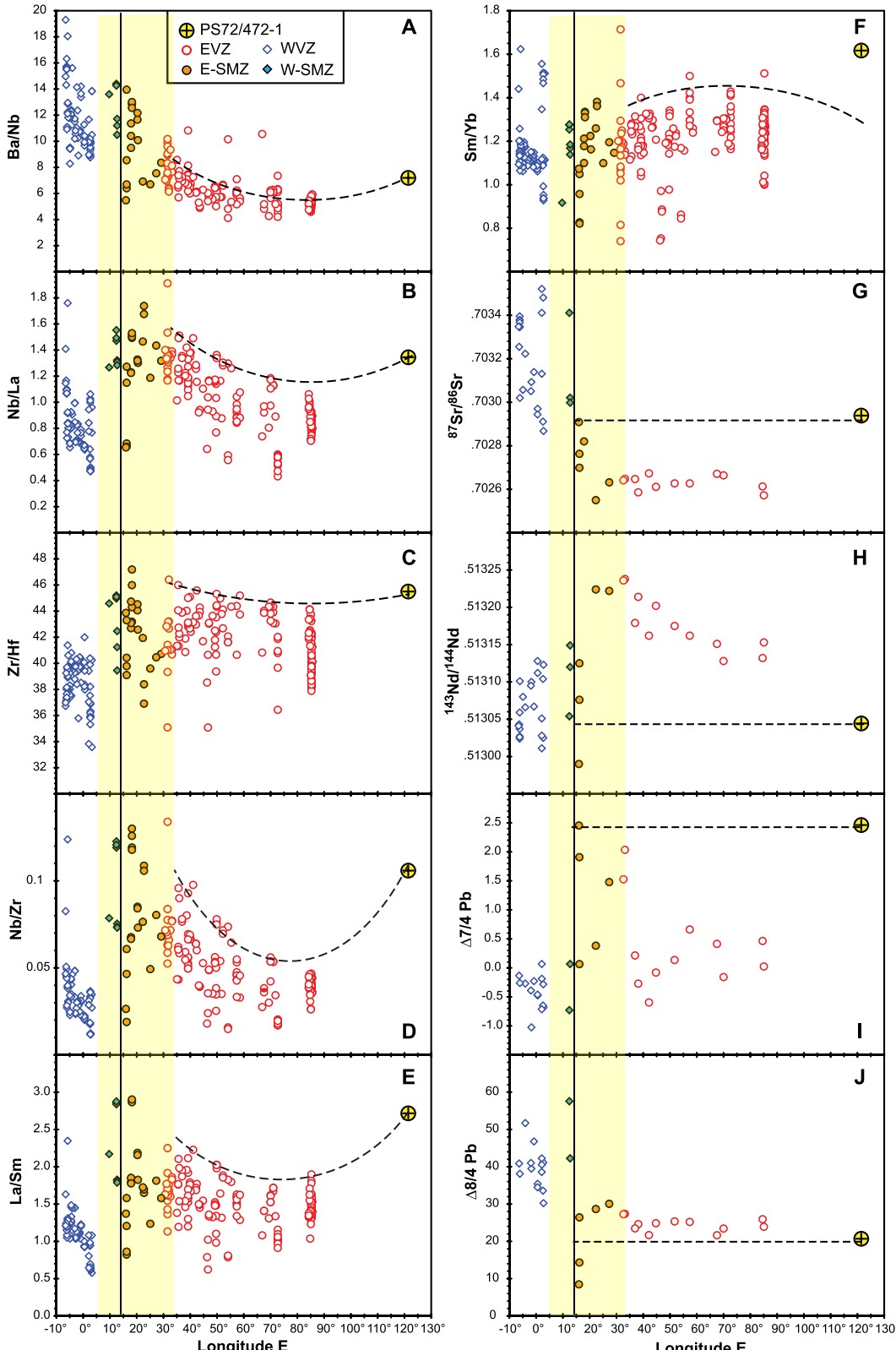

**Fig. 7 Geochemical trends along the Gakkel Ridge.** Along-axis variations of incompatible element ratios of **A** Ba/Nb, **B** Nb/La, **C** Zr/Hf, **D** Nb/Zr, **E** La/Sm, and **F** Sm/Yb in Gakkel lavas and long-axis variations of **G** $^{87}Sr/^{86}Sr$, **H** $^{143}Nd/^{144}Nd$, and Pb isotopes expressed as **I** Δ7/4Pb and **J** Δ8/4. The Delta (Δ) notation reflects deviations from typical compositions of Pacific and Atlantic MORB on $^{208}Pb/^{204}Pb–^{206}Pb/^{204}Pb$ diagrams[28]. Note that the majority of Gakkel melts are tholeiitic, while alkaline melts are infrequent but occur in each segment[10]. The vertical line marks the isotopic boundary at ~14°E[18]. The dashed line shows maximum incompatible element enrichment and variation in $^{87}Sr/^{86}Sr$, $^{143}Nd/^{144}Nd$, and Pb isotopes discussed in the text. The figure is adapted from ref. [10], their Fig. 5 and is based on published data of volcanic glasses (compilation of ref. [58]) and melt inclusions (refs. [32,59]) and a basalt sample from the GRD[10].

the EVZ side of this 14°E boundary increase to higher levels of enrichment compared to the rest of the EVZ, most clearly in Δ7/4 Pb. This observation is consistent with a greater proportion of low-degree melts from the underlying DUPAL and EVZ mantle sources due to thicker lithosphere and/or cooler mantle. However, the transition from very thin to the thick crust at the SMZ-EVZ boundary[14,19] is expected to have a spike in REE enrichment reflecting the thicker SMZ lid. The EVZ trace element geochemical trends are consistent with decreasing % melting of a variably enriched (metasomatized) MORB (EVZ-type) source linked to a thermal anomaly that grades to cooler temperature away from roughly the middle of the EVZ at roughly 70–80°E. In the more magmatic WVZ, geochemical trends are consistent with a thermal anomaly fed by DUPAL-like mantle inflowing from the North Atlantic[18]. In conclusion, adopting the interpretations of the most recent tomographic model for the Arctic published in 2017, we argue that our new [40]Ar/[39]Ar data for the Gakkel Ridge shows that spreading rate and accretion style correlates with locations of thermochemical anomalies in the asthenosphere beneath the ridge. Evidence that the structure of the lithosphere, the extent of magmatism, and its composition correlate with spreading rate and style of accretion also links them in turn to mantle temperature.

Next, we consider how the temperature structure of the underlying mantle might be controlling variations in spreading rate and accretion style, composition, and thickness of the lithosphere (depth). Ridge-push and slab-pull are considered to be the processes responsible for driving spreading at ridges. Ridge-push is the result of gravitational forces acting on the young, raised oceanic lithosphere forming at ridges[33]. In plates without any subduction, such as the Arctic region, ridge-push might be the main or even the only force driving spreading and plate motion[34,35]. Thus, small changes in the temperature of the asthenosphere and, to a much lesser extent, composition due to thermochemical anomalies[36], can lead to large variations in the volume of melts in the melting zone and the corresponding elevation of spreading ridges[26,27,37,38]. Heat at a ridge will also thin the lithosphere by weakening the lithosphere closer to the surface and elevating the depth boundary between the brittle lithosphere and the weaker, ductile asthenosphere[39]. For example, at the Gakkel Ridge, earthquakes show thinning of the lithosphere to ≤15 km below volcanoes (or sites of basalt exposure) in the magmatic sections, compared to thick lithosphere (≤40 km) (with thin or absent oceanic crust[2,3]) below amagmatic sections[14,19,40]. Slightly increasing mantle temperatures (and a slight lowering of lateral densities) will also destabilise the normal thermal and compositional stratification leading to the development of gravitational instabilities or small-scale convection (SSC), e.g., refs. [41,42]. Ample observational data is showing that active (buoyant, three-dimensional, diapiric, episodic) mantle upwelling distinguishes slow-spreading ridges from intermediate- and fast-spreading ridges[43–49]. Boundary-layer instabilities will further thin/erode the base of the cold, thick Arctic lithosphere[14,19], especially under ultraslow moving lithosphere[41,42]. In summary, we argue here that hotter mantle associated with the low-velocity anomalies under the EVZ and WVZ leads to a thinner, more elevated lithosphere that increases the gravitational forces acting on the young, raised oceanic lithosphere.

On a global scale, variations in depths (uplift of the lithosphere) of the ocean ridges and lava composition are considered to reflect largely the temperature structure of the underlying mantle[26,27]. Gakkel is the deepest of the ocean ridges and provides a test of this global correlation and its origin[50]. Here we show that there is a relation between the temperature structure of the underlying mantle and [40]Ar/[39]Ar-defined spreading rates and accretion modes and by inference the thickness of the lithosphere

(depth) and ridge geochemistry. We argue that this is evidence that the global relation between the depths of the ocean ridges, lava composition, and temperature[26,27] is valid for most ridges. Moreover, it implies that global models can be expanded to include spreading rate and style of crustal accretion.

We have shown also that [40]Ar/[39]Ar age data can robustly measure spreading rates along spreading centres, even along an ultraslow spreading ridge. Isotopically dating young, low potassium spreading ridge samples is now possible thanks to recent advances in high-resolution multi-collector mass spectrometry and sample preparation methods. The most significant original insight provided by these dates is that, together with the latest tomography model for the Arctic[17], they show that steady spreading rate correlates with hotter regions in the underlying upper mantle. Whereas amagmatism reflects colder underlying upper mantle. Our findings imply that ultra-slow spreading is not necessarily a linear process with constantly decreasing spreading rates towards the rotation pole, as predicted by global models assuming rigid tectonic plates and extrapolation from older magnetic anomalies at the Gakkel Ridge. Rather, ultra-slow spreading is discontinuous with several segments acting independently from each other depending on the (changing) thermal structure under the ridge.

## Methods

**[40]Ar/[39]Ar dating of Gakkel Ridge samples.** The groundmass samples were prepared following the methods of ref. [51]. The 200–180 μm samples measured at Oregon State were cleaned in a series of hour-long acid baths, progressing from 1N HCl to 6N HCl to 1N HNO$_3$ to 3N HNO$_3$, followed by a final milli-Q water bath. Each separate was picked by hand under a binocular microscope to ensure the removal of alteration, and to confirm the purity of the separate. Groundmass samples were irradiated for 6 hours in the CLICIT position at the Oregon State University TRIGA reactor. Incremental heating experiments were conducted for each sample. Irradiated samples were loaded into copper planchettes for analysis using a Thermo Scientific ARGUS-VI multi-collector mass spectrometer at the OSU Argon Geochronology Laboratory following the procedure described in ref. [52]. All ages are calculated relative to Fish Canyon Tuff (FCT) sanidine with an age of 28.201 Ma[53] and using the decay constants after ref. [54]. The correction factors for neutron interference reactions at the TRIGA are $(2.64 \pm 0.02) \times 10^{-4}$ for ([36]Ar/[37]Ar)Ca, $(6.73 \pm 0.04) \times 10^{-4}$ for ([39]Ar/[37]Ar)Ca, $(1.21 \pm 0.003) \times 10^{-2}$ for ([38]Ar/[39]Ar)K and $(8.6 \pm 0.7) \times 10^{-4}$ for ([40]Ar/[39]Ar)K. Ages were calculated using the ArArCALC v2.7.052 software of ref. [55], with errors including uncertainties on the blank corrections, irradiation constants, J-curve, collector calibrations, mass fractionation, and the decay of [37]Ar and [39]Ar.

The quality of an Ar/[39]Ar step-heating experiment is assessed based on the following criteria: an acceptable age plateau (1) includes at least 50% of the gas released, (2) has a mean square weighted deviation (MSWD) of ~1.0 and within the statistically allowed upper limit, (3) shows an inverse isochron with a [40]Ar/[36]Ar intercept of about 295.5 ± 2σ, and (4) has a concordant plateau, isochron, and total fusion ages and (5) a *p*-value >5%. Seven samples have a plateau with <50% 30Ar and/or a *p*-value <5% (see Table 1 and Fig. 1). Full analytical results are available in Supplementary Dataset 2.

**Calculation of spreading rates.** Spreading rates are calculated using dated samples that have been dredged from magmatic structures that have been fragmented apart and transported away as seafloor spreads from the active central rift axis. Full spreading rates are calculated based on the separation distance between these magmatic fragments and measured isotopic ages: SR = distance (km)/age (Ma).

In an ideal scenario spreading rates could be calculated using isotopic ages for samples from in situ fragments of the same magmatic edifice located on the opposite sides of the rift axis and at the same depth. But the in situ location of dredged samples is not known reliably and the depths of dredged samples on the opposite side of the axis might be different. For example, basalt might have fallen from higher up in the rift valley wall, or flowed a significant distance vertically and/or horizontally (parallel-to-spreading direction) after eruption. Dated samples are not available necessarily from directly opposite sides of the rift axis or fragments of the same magmatic edifice. In the case of the SMZ low volume of melts can reach the surface away from the rift axis via large faults at off-axis distances of up to 10–15 km.

Magmatic fragments should be rifted completely and then transported orthogonally away from the rift axis by symmetric seafloor spreading. Asymmetric spreading is evident from a comparison of half spreading rates (samples site distances from the rift axis) and full spreading rates (age-distance profiles) (Table 1). This implies that rift zones might have migrated/jumped such that the rifts extend across a wide zone. Magmatic structures are not necessarily

dismembered symmetrically or only partially. Moreover, young magmatic centres can be in the process of breaking apart.

Another consideration is that the walls of the rifted and transported magmatic fragments should be vertical. If one or both walls facing the rift axis is sloping due to, for example, up-faulting this will lead to vertical variation in the calculated spreading rate. In such cases calculating profile lengths from roughly the middle of the rift, walls can average out this effect.

Depth profiles across the ridge should be mapped completely by high-resolution multibeam (100-m contour) mapping rather than the much lower resolution International Bathymetric Chart of the Arctic Ocean (IBCAO)[56]. This is not always the case so depth profiles are sometimes oblique to the spreading ridge to identify and connect fragments on opposite sides of the rift axis.

But the main source of uncertainty that might bias systematically the calculation of spreading rates rises from the isotopic dates and the navigation of dredge positions and the length of the age profile. Using an assumed uncertainty of 0.5 km for the sample locations/age-profiles and the analytical uncertainty reported for the isotopic ages we have calculated the minimum and maximum spreading rates for each sample and then determined via linear regression a spreading rate of 11.1 ± 0.9 for WVZ/EVZ and a rate of 7.6 ± 0.5 for SMZ/GRD.

In summary, due to the difficulty in quantifying the uncertainty for individual spreading rates we base our discussions and findings on the average spreading rates for the different ridge segments (Fig. 1).

## Data availability
The authors declare that all the data for the $^{40}Ar/^{39}Ar$ age determinations and geochemical analyses supporting the findings of this study are available within the paper and its supplementary information files.

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

## Acknowledgements

P.J.M. acknowledges financial support from US National Science Foundation Grants OPP 9911795 and PLR 0425892 and the University of Tulsa McMan Endowment.

## Author contributions

W.J. designed this project. P.J.M., W.J. and M.C.S. participated in the AMORE expedition. P.J.M. provided the sample material. D.P.M. and A.A.P.K. carried out the $^{40}Ar/^{39}Ar$ age determinations. M.C.S. produced Fig. 1 and helped J.M.O. calculate the spreading rates. J.M.O. wrote the original draft and all authors reviewed the manuscript and contributed to the discussion in this paper.

## Competing interests

The authors declare no competing interests.
