## [Peer Review File · Nature Communications]

REVIEWER COMMENTS

Reviewer #1 (Remarks to the Author):

This paper presents a lot of new $^{40}\text{Ar}/^{39}\text{Ar}$ data on dredge samples from the Gakkel ridge. Most data are of good quality and bring precise constraint on the rate of eruptions of various section of the ridge. The authors then combine those data with published geochemical and geophysical data to attempt to understand the functioning of a slow spreading ridge. The topic is interesting and would fits well in NatCom. At this stage, I have some level reservation on several of the conclusions reached by the authors (correlation rate with geochemistry or magmatic cyclicality) as these are not (yet?) supported by the data (nor positively or negatively) and I recommend a bit more work to strengthen some of the arguments, by some more numerical demonstrations, if this is possible at all (cf. comment below). Since the topic is very interesting, but in the light of my comments below, I'm currently recommending major revisions (or rejection), with another critical round of reviews after that. That should give the authors the chance to strengthen their arguments.

Cf. attached file with detailed comments

This paper presents a lot of new $^{40}\text{Ar}/^{39}\text{Ar}$ data on dredge samples from the Gakkel ridge. Most data are of good quality and bring precise constraint on the rate of eruptions of various section of the ridge. The authors then combine those data with published geochemical and geophysical data to attempt to understand the functioning of a slow spreading ridge. The topic is interesting and would fits well in NatCom. At this stage, I have some level reservation on several of the conclusions reached by the authors (correlation rate with geochemistry or magmatic cyclicality) as these are not (yet?) supported by the data (nor positively or negatively) and I recommend a bit more work to strengthen some of the arguments, by some more numerical demonstrations, if this is possible at all (cf. comment below). Since the topic is very interesting, but in the light of my comments below, I'm currently recommending major revisions (or rejection), with another critical round of reviews after that. That should give the authors the chance to strengthen their arguments.

Detailed comments:

Abstract "*Furthermore, our dating shows that variations in spreading rates, structure of the lithosphere, and the extent of magmatism and geochemistry, correlate with the locations of thermochemical anomalies in the asthenosphere beneath the ridge.*" That is not really how I read the text. Rather I read "Furthermore, our dating shows that variations in spreading rates correlate with the structure of the lithosphere, the extent of magmatism and geochemistry and the locations of thermochemical anomalies in the asthenosphere beneath the ridge" The distinction is in fact important because the new information here is the new age data which can then be compared to those other things.

L75-77: the new rates seems very well supported by the data. Very nice. That being said, it is not really acceptable to provide such numbers without uncertainties (is it 11.1 ± 0.1 or ± 4 mm/yr?) and without probability or Chi^2 test values to show how robust they really are (R^2 values would do).

Ar data (Annex 2): The vast majority of the data are really excellent. However, a few of them (4 of them from a quick brush) have been overinterpreted. Some of these data are not really ages at all, they are error ages at best with some data have less than 50% of ^{39}Ar included extracted from a structured age spectrum which rather suggest alteration:

Several samples have P-value below the acceptable threshold making them scatter (error) ages. That's especially important, since these data are based on groundmass sample which are notorious to give *plateau* ages which are nevertheless discordant with more accurate mineral

separate ages from the same rock. That's the age that was used in the rate calculation as per Extended table Table 1. Although I note a concordant inv isochron for this particular case:

Results	$^{40}\text{Ar}/^{36}\text{Ar} \pm 2\sigma$	$^{40}\text{Ar}/^{39}\text{Ar}(\text{k}) \pm 2\sigma$	Age $\pm 2\sigma$ (ka)	MSWD	$^{39}\text{Ar}(\text{k})$ (%,n)	K/Ca $\pm 2\sigma$
Age Plateau		0.26651 ± 0.02231	791.6 ± 66.3	1.83	88.28	0.0122 ± 0.0050
Error Mean		$\pm 8.37\%$	$\pm 8.37\%$	1%	20	
		Full External Error ± 68.6		1.65	Confidence Limit	
		Analytical Error ± 66.3		1.3526	Error Magnification	

Table 1 must include the P-value (\pm MSWD) for each of the ages. Error ages should be flagged in there (low amount of ^{39}Ar , or low P-value) for who ever want to use the data in the table. I also urge the authors to flag those samples in their Fig. 1 and 3. Those 4 error ages are not very robust so the reader should be informed within the main text for which of those data this is the case. I would recommend using semi-transparent symbols, to distinguish those from true plateau ages.

L102-103: The WVZ is different in term of composition (especially in ^{208}Pb), however, the difference between the SMZ and EVZ is much less obvious, despite the vast difference in spreading rate. With the overwhelming majority of the data clearly overlapping in La/Sm (a proxy for the amount of partial melting), 7/4 and 8/4. In particular, only four data points stand out in the La/Sm which suggest a lower degree of melting for those basalts, but all the other one have comparable degree of melting (in the red box below, which could have been extended to even the WVZ). So the difference in Geochem does not seem to play a huge role, except for the four outliers. Does the $^{40}\text{Ar}/^{39}\text{Ar}$ ages come from these four rocks? There are only four ages to define the 8 mm/yr trend after all. If not, then the Geochem is not related to the degree of melting, at least based on the graphs provided.

So at this stage I'm seeing two different speeds WVZ + EVZ against a slower SMZ (GRD has only one datum, so it's hard to really conclude anything), whereas the La/Sm and 8/4 geochemical ratio rather seems to group the SMZ and EVZ against the WVZ. Note that this might have been clearly demonstrated in previous papers, which I did not check, but regardless, such a demonstration should be clearly provided in this paper as well since part of conclusion is based on that.

L104-106: as such, a correlation between spreading rate and geochemistry, is not well established at this stage, and would require a bit more work (maybe using different ratios, or partial melting models?). Perhaps mantle source mixing?

L110: what I'm personally seeing is a continuous type of magmatism without true time clusters. There are really too few data to define magmatic cycles as per Fig. 3a. Probability density diagram like that are dependent on the choice of the bins which can influence how many peak one can see. Therefore, looking the individual ages (yellow dots of Fig. 3a), I would rather argue for a very continuous type of magmatism. But it's not about what the author think they visually see, or what I think I visually see – it's about true numbers. What they author would need to do, is to estimate the probability that their “grouping” is not due to chance associated with sample bias.

For example, this graph was crated using the RAND function on Excel (generating random number between 0 and 3.8) using 26 data points. Those show artificial group relatively similar to Fig. 3a (and b).

So while I'm NOT saying that there can't be any groups in what the authors shown, such a validating those grouping would require a bit more of a demonstration, preferably using some sort of statistical likelihood calculations, or perhaps correlation with mag anomalies, if any. I personally think that what we are seeing is not groups, but rather the gaps (i.e., absence of data) between an otherwise continuous dataset due to sample bias. Right or wrong, this has to be better demonstrated. This of course, is not helped by the fact that those are dredge rocks.

L125 – period of 500 ka. Notwithstanding my remark above, I wonder how this value is calculated? It is really not apparent from Fig. 3 (nor there is a lot of support for such a value). The authors needs to give more details on their calculations.

L165 – if the mantle melts at 150 km, then one should be able to see garnet bearing mantle signature, which is notoriously visible in the fractionation of REE ratios. Since it is part of the conclusion, such demonstration must be established as well. This is especially true in the case of low degree of partial melting of the MORB mantle since its REE composition will not mask the REE composition from a deeper melt source.

That being said, the correlation between the thermal anomaly at 150 km and the spreading rate (Fig. 4a), is much more convincing than the geochemical correlation. Note that I don't really understand Fig. 4b though, I can't seem to really pinpoint where the Gakkel ridge is on those two velocity graphs.

I found the discussion hard to follow, with a succession of seemingly poorly connected arguments (they are not unrelated at all of course, but it is hard to grasp without some efforts as the transitions are abrupt). This is in contrast to the abstract and conclusions which are much more clear on what is being demonstrated and tested. I'm sure it all makes sense to the authors, of course, but when it is the first time that one read this paper, parts will appear confusing. For example, the part about the deeper source (Cf. remark above). What are the implications of that? as it stands, it's just an observation, but deep mantle contribution to ridge is potentially a very important observation – that should be developed better to give the reader a sense of why this is important. I suggest rewriting part of the discussion to make it flow better and perhaps keeping it a tad more simple.

Lots of the conclusions come from geochemical and geophysical data that have been published and are combined with the geochronology part. It is not like geochron really enlighten some of those data (the slow and fast rates where known before “*~13 mm/yr in the west to only ~6 mm/yr in the east*”). So my question is did we learn of the much lower rate in the SMZ in this study, or was that known before as well? If new, I recommend the authors to sell this point better. Regardless, these data are important because they provide a much more accurate values for the two different spreading rates between WVZ and EVZ vs. SMZ and GRD which will be used by readers in the future. To me the more important conclusion is spreading rate \approx thermal/compositional anomaly of a deeper mantle source.

Reviewer #2 (Remarks to the Author):

The manuscript by O'Connor et al. presents an impressive dataset of $^{40}\text{Ar}/^{39}\text{Ar}$ dates from the Gakkel Ridge. The authors apply this dataset to determine spreading rates along the ridge, then compare the spreading rates to previously documented chemical variations and larger-scale mantle velocity anomalies.

This is an interesting and thought-provoking dataset and should definitely be published in some form (although not necessarily in Nature Communications). I have some concerns and questions regarding the current manuscript, which I outline below:

1. I wonder whether there truly are differences in spreading rate at the WVZ-EVZ versus SMZ-GRD, or whether they simply reflect different styles of crustal accretion. The key to this question appears to be the data from the SMZ. The authors report 10 new dates from this segment. They discard half of these datapoints, which yield robust younger dates, but that the authors argue "do not yield realistic spreading rates due to their mode of emplacement". The authors argue the younger dates reflect eruption of lavas off axis due to magma transport along faults. This seems to be a reasonable interpretation, although I note that it is not supported by evidence beyond the young dates for these lavas, given that these are dredge samples and the context of the samples is not known.

Excluding the younger dates, a linear fit of the SMZ data suggest a slightly slower spreading rate (7.6 mm/yr) than the WVZ and EVZ segments (11.1 mm/yr). I wonder whether this lower rate is robust or whether it instead reflects a wider zone of magmatic accretion along the SMZ. The excluded younger dates suggest there is significant off-axis magmatism (half the dated samples). If magmas are intruded over a wider zone, lavas erupted at the ridge axis will appear to move off axis more slowly, as some spreading is taken up by off-axis intrusions. In other words, it seems possible that the WVZ and EVZ are characterized by focused magmatism along the ridge axis, whereas along the SMZ magmatism is distributed across a wider area. At the WVZ and EVZ the focused magmatism along the ridge would lead to the expected increase in dates moving away from the ridge axis. In contrast, at the SMZ, the more diffuse zone of magmatism would lead to some younger dates off axis (as observed in the data). The "apparent" spreading rate for samples erupted on axis would also be slower, because it will take longer for lavas erupted on-axis to move off axis if spreading is accommodated over a wider zone. The same could be true for the GRD, which has only a single dated sample.

The spreading rates calculated using the younger dates from the SMZ should be included on the inset in the lower right of Figure 1. The SMZ data would then form a cloud of data at both slower and faster spreading rates than the WVZ and EVZ.

Whether or not there are differences in spreading rate between the WVZ-EVZ and SMZ-GRD is important, as much of the manuscript focuses on correlating the variations in spreading rate to other parameters (e.g., chemistry, mantle seismic velocity).

2. The authors argue that the dates record cyclic magmatic trends (Figure 3). This is highly speculative, is not supported by the current data and I would argue this section should be cut from the manuscript. The authors base their argument on a plot of frequency versus date. On this plot, they note that in some cases they have a single date from a given age range (e.g. 0.1 Ma), while at other times they have two samples that give the same date (e.g., 0.3 Ma)—three samples happen to yield a date of ~ 1.7 Ma. The authors attribute the frequency of a given date to reflect the volume of magmatic activity, where periods with two dated samples reflect increased magmatism compared to periods with zero or one dated sample.

The number of dated samples from a large geographic area is simply too low to assess temporal variations in the volume of erupted magmas. The apparent "peaks" of magmatic activity may reflect increased magmatic activity, but they may also just reflect random sampling of a uniform age distribution; I consider the latter to be more likely in this case (i.e., the authors happened to date two rocks with an age of 0.3 Ma and one rock with an age of 0.5 Ma). To quantitatively assess temporal variations in the volume of

magmatism, the authors would need to date a much larger number of samples, and ideally systematically sample a set geographic area.

Related to this, I note that dates from individual "magmatic cycles" that the authors define and plot in figure 1b, do not all actually overlap within uncertainty (e.g., only two of the three dates in cycle 2 overlap within uncertainty). Other "magmatic cycles" are defined by multiple dates with high uncertainties, where the overlap may reflect the larger analytical uncertainties, rather than a pulse in magmatism (e.g., cycle 5).

3. The authors argue that the distinct spreading rates and chemistry of the WVZ, EVZ, SMZ and GRD may be related to previously observed low shear wave velocity zones, which have been interpreted to reflect elevated asthenospheric temperatures. They suggest that low velocity (higher T) zones below the EVZ and WVZ lead to higher degrees of melting, more robust magmatism and faster spreading. This does not appear to be supported by the distribution of the low velocity anomalies. Notably, the low velocity anomalies at 150 km underlies the EVZ and WVZ, but also extend below the dated sample from the GRD and several samples from the SMZ that are adjacent to the EVZ. The GRD and SMZ samples are interpreted to record lower spreading rates, but there is no offset in spreading rate between SMZ samples underlain by the low velocity zone and the other SMZ samples, as would be expected if the presence of the low velocity zone leads to higher spreading rates.

The correlation between spreading rate and chemistry appears to be more robust (although I have questions about whether there actually are differences in spreading rates between the segments, as discussed above).

4. One of the main conclusions of the paper is "that (slight) differences in mantle temperature/magma supply causes certain segments to spread with slightly different velocities." The authors do not discuss why this might be or what physical process would lead to different spreading rates at segments with higher magma supply? If spreading rate is controlled by large scale plate movements, it is not clear to me how or why variations in magma supply would impact the spreading rate. This does not seem sustainable over longer timescales. How can the EVZ and WVZ spread at a higher rate than the SMZ? If the plates are spreading at 11.1 mm/yr, how can the SMZ spread at 7 mm/yr over long timescales?

5. The main contribution of this manuscript is an impressive new $^{40}\text{Ar}/^{39}\text{Ar}$ dataset from the Gakkel Ridge. The authors briefly describe the dating method and do include complete data reports for all the dated samples in the Supplementary material, but there is no real discussion in the main or supplemental text of the $^{40}\text{Ar}/^{39}\text{Ar}$ results. Given that this is the heart of the paper, the authors should include a discussion of the quality of the results, systematics of the $^{40}\text{Ar}/^{39}\text{Ar}$ spectra, etc. Related to this, it is surprising to me that Anthony Koppers is the last author on the paper, when the main contribution of the manuscript is the new $^{40}\text{Ar}/^{39}\text{Ar}$ dates, which were collected in his lab.

Given the importance of the $^{40}\text{Ar}/^{39}\text{Ar}$ dates in this manuscript, it should be reviewed by a 40-39 specialist (which I am not).

Overall, the new contribution of this manuscript is the large ^{40}Ar - ^{39}Ar dataset. The authors argue that the dataset suggests there are differences in spreading rate between the EVZ-WVZ and SMZ-GRD. As I outline above, I was not fully convinced by this interpretation and the authors do not discuss a physical basis for why the SMZ-GRD have a lower spreading rate and how that is accommodated with respect to large-scale plate motions.

Some of the main conclusion of the manuscript seem disconnected from the new data presented in the paper (i.e., the $^{40}\text{Ar}/^{39}\text{Ar}$ dates). For example, at the end of the abstract, the authors state: "We conclude that relatively slight changes in mantle temperature and composition (volatiles) are likely to cause large variations in the amounts of partial melting and the vigour of boundary layer instabilities, which in turn determines the volume of melt reaching the surface through the Arctic oceanic lithosphere. We conclude that magmatic and amagmatic seafloor generation in ultra-slow spreading centers reflects the distribution of thermochemical anomalies in the upper mantle." These are both interesting conclusions, but are based on

previously published geochemical and seismic data, rather the results of this study. If there are changes in spreading rate between the different ridge sections, they may be related to the geochemical and seismic observations; however, the authors do not discuss how these factors could explain or lead to variable spreading rates, beyond noting that the changes in spreading rate appear to correlate with the geochemical and (to a lesser extent) seismic signals.

Detailed comments:

Lines 16 to 21: These two sentences both start with "We conclude". Reword?

Lines 42 to 44: "Furthermore, crustal thickness does not become thinner as spreading rate decreases along Gakkel Ridge from ~ 13 mm/yr in the west to only ~ 6 mm/yr in the east at 120E Karasik, 1968; Vogt et al. 1979; Jokat and Schmidt-Aursch 2007 and references therein; Schmidt-Aursch and Jokat, 2016."

The data presented in this manuscript show that spreading rate does not decrease along the ridge. It is confusing to present this as fact, when it is actually disproven by the results of the paper.

Line 66 and throughout: Many in the geochronologists distinguish between date and age. A date is a measured value while an age is the true timing of a geologic event (e.g., The 75 Ma date is interpreted as the age of pluton crystallization). The authors might consider adopting this convention throughout the manuscript.

Lines 85 to 87: "Here we like to consider whether there is a causal relation between spreading rates calculated with the new isotopic ages and the various unexpected findings arising from the AMORE expedition discussed already."

Consider removing "we like" and replacing "unexpected" with unexpected.

Lines 89–91: How are the proposed strike slip faults different from transform faults?

Figure 1: It would be useful to label the dotted line in the spreading rate panel (second from top; e.g., "NNR-MORVEL56 global model")

The excluded younger dates from SMZ should also be included on the Age vs. Profile plot in the lower right.

Figure 1 caption, line 364: Typo., replace "expacement" with emplacement.

Figure 3b. It took me a bit to understand what this plot represents. It is well described in the caption, but the authors should add a label to the y-axis. What criteria was used to define the "magmatic episodes?"

REVIEWER COMMENTS

We thank both reviewers for their very constructive and insightful reviews.

Reviewer #1 (Remarks to the Author):

This paper presents a lot of new $^{40}\text{Ar}/^{39}\text{Ar}$ data on dredge samples from the Gakkel ridge. Most data are of good quality and bring precise constraint on the rate of eruptions of various section of the ridge.

By the way, we now use the above phrasing in our revised text as follows: “...provide precise constraint on the spatial and temporal distribution of volcanic eruptions at various sections of the ridge”

The authors then combine those data with published geochemical and geophysical data to attempt to understand the functioning of a slow spreading ridge. The topic is interesting and would fits well in NatCom. At this stage, I have some level reservation on several of the conclusions reached by the authors (correlation rate with geochemistry or magmatic cyclicality) as these are not (yet?) supported by the data (nor positively or negatively) and I recommend a bit more work to strengthen some of the arguments, by some more numerical demonstrations, if this is possible at all (cf. comment below). Since the topic is very interesting, but in the light of my comments below, I'm currently recommending major revisions (or rejection), with another critical round of reviews after that.

That should give the authors the chance to strengthen their arguments.

We appreciate very much the reviewer's constructive approach.

Detailed comments:

Abstract “Furthermore, our dating shows that variations in spreading rates, structure of the lithosphere, and the extent of magmatism and geochemistry, correlate with the locations of thermochemical anomalies in the asthenosphere beneath the ridge.” That is not really how I read the text. Rather I read “Furthermore, our dating shows that variations in spreading rates correlate with the structure of the lithosphere, the extent of magmatism and geochemistry and the locations of thermochemical anomalies in the asthenosphere beneath the ridge” The distinction is in fact important because the new information here is the new age data which can then be compared to those other things.

A very good point. We have moved and corrected the text accordingly.

L75-77: the new rates seem very well supported by the data. Very nice. That being said, it is not really acceptable to provide such numbers without uncertainties (is it 11.1 ± 0.1 or ± 4 mm/yr?) and without probability or Chi^2 test values to show how robust they really are (R^2 values would do).

We now report uncertainties for the spreading rates as explained in the Methods Section as follows:

“But the main sources of uncertainty that might bias systematically the calculation of spreading rates [$\text{SR} = \text{distance (km)}/\text{age (Ma)}$] arise from the isotopic dates and the navigation of dredge positions and the length of the age-profile. Using an assumed uncertainty of 0.5 km for the sample locations/age-profiles and the analytical uncertainty reported for the isotopic ages we have calculated the minimum and maximum spreading rates for each sample and then determined via linear regression a spreading rate of 11.1 ± 0.9 for WVZ/EVZ and a rate of 7.6 ± 0.5 for SMZ/GRD. The results are shown in Supplementary Figure 3 and Supplementary Table X). Whereas regressing the age data and age-profiles to assess the scatter of the samples by yields residuals ranging from -0.63 to 1.31 for SMZ/GRD and -0.91 to 0.99 for WVZ/EVZ.”

Ar data (Annex 2): The vast majority of the data are really excellent. However, a few of them (4 of them from a quick brush) have been overinterpreted. Some of these data are not really

ages at all, they are error ages at best with some data have less than 50% of ^{39}Ar included extracted from a structured age spectrum which rather suggest alteration:

Several samples have P -value below the acceptable threshold making them scatter (error) ages. That's especially important, since these data are based on groundmass sample which are notorious to give plateau ages which are nevertheless discordant with more accurate min separate ages from the same rock. That's the age that was used in the rate calculation as per Extended table Table 1. Although I note a concordant inv isochron for this particular case:

Results	40(a)/36(a) $\pm 2\sigma$	40(r)/39(k) $\pm 2\sigma$	Age $\pm 2\sigma$ (ka)	MSWD	39Ar(k) (%,n)	K/Ca $\pm 2\sigma$
Age Plateau		0.26651 \pm 0.02231	791.6 \pm 66.3	1.83	88.28	0.0122 \pm 0.0050
Error Mean		\pm 8.37%	\pm 8.37%	1%	20	
		Full External Error \pm 68.6	Analytical Error \pm 66.3	1.65	2	Confidence Limit
				1.3526		Error Magnification

Table 1 must include the P -value (\pm MSWD) for each of the ages. Error ages should be flagged in there (low amount of ^{39}Ar , or low P -value) for whoever want to use the data in the table. I also urge the authors to flag those samples in their Fig. 1 and 3.

We have now included MSWD and P -value for each of the ages in Table 1.

Those 4 error ages are not very robust so the reader should be informed within the main text for which of those data this is the case. I would recommend using semi-transparent symbols, to distinguish those from true plateau ages.

We have added the following to the methods section:

“A plateau date of 1.65 ± 0.09 Ma for PS59-226-23 is based on 45% of the released ^{39}Ar . However, the inverse isochron age is within analytical error at 2.44 ± 0.58 Ma and the total fusion date of 1.59 ± 0.05 Ma agrees. Five other samples have P -values less than 5% however they all have concordant isochrons and total fusion ages. One of these samples is attributed to fault-related volcanism in the SMZ whereas the other four are equally divided between the WVZ and EVZ. None are used in calculating the average spreading rate in the SMZ. Full analytical results are available in Supplementary Dataset 2.”

And the following to the main text:

“The criteria for assessing the quality of the age data are discussed in the methods section. Although six samples do not meet all criteria (Table 1), they still provide robust estimates of the eruption ages as each of them shows concordant plateau, isochron, and total fusion ages; however, we have not used these six lesser-quality ages in our calculations of the slower average spreading rate in the SMZ that we are discussing in the following section, because their inclusion/exclusion doesn't change the outcome of our analyses.” While using semi-transparent symbols is a good idea we think that this would confuse the non-expert reader. But of course, it's not an issue to make the symbols transparent as might be deemed necessary.

L102-103: The WVZ is different in term of composition (especially in ^{208}Pb), however, the difference between the SMZ and EVZ is much less obvious, despite the vast difference in

spreading rate. With the overwhelming majority of the data clearly overlapping in La/Sm (a proxy for the amount of partial melting), 7/4 and 8/4. In particular, only four data points stand out in the La/Sm which suggest a lower degree of melting for those basalts, but all the other one have comparable degree of melting (in the red box below, which could have been extended to even the WVZ).

So the difference in Geochem does not seem to play a huge role, except for the four outliers.

Does the $^{40}\text{Ar}/^{39}\text{Ar}$ ages come from these four rocks?

The figure showing the La/Sm ratios below and the new geochemistry Fig. 7 are adapted from a figure in ref. Jokat et al., 2019, which is based on published data of volcanic glasses (compilation of Gale et al., 2013) and melt inclusions (Shaw et al., 2010; Wanless et al., 2014) and a basalt sample from the GRD Jokat et al., 2019.

The outlier SMZ samples are from the compilation of Gale et al (an Excel file is available for download via a link in Gale et al., 2013). The dated samples (Table 1) include the following La/Sm outliers: HLY0102-036-009 ('near rift axis'); PS59/252-1-001 ('fault related'); and PS59/312-011 (used to calculate SMZ spreading rate). The compilation also includes analyses from the same samples (PS59-252-001WS; PS59-252-SG2) and the same dredge haul (HLY0102-036-001).

There are only four ages to define the 8 mm/yr trend after all. If not, then the Geochem is not related to the degree of melting, at least based on the graphs provided.

So at this stage I'm seeing two different speeds WVZ + EVZ against a slower SMZ (GRD has only one datum, so it's hard to really conclude anything), whereas the La/Sm and 8/4 geochemical ratio rather seems to group the SMZ and EVZ against the WVZ. Note that this might have been clearly demonstrated in previous papers, which I did not check, but regardless, such a demonstration should be clearly provided in this paper as well since part of conclusion is based on that.

L104-106: as such, a correlation between spreading rate and geochemistry, is not well established at this stage, and would require a bit more work (maybe using different ratios, or partial melting models?). Perhaps mantle source mixing?

So the difference in Geochem does not seem to play a huge role, except for the four outliers.

We address the issue only four outliers marked in the figure above by including additional graphs showing along axis variations in Ba/Nb, Nb/La, Zr/Hf, Nb/Zr and Sm/Yb (please see our new Figure 7). Moreover, instead of La/Sm we now use/cite the positive correlation between Na_{8.0} (Na₂O normalized to 8% MgO) and depth as a proxy for % of melting Klein and Langmuir, 1987; Langmuir et al., 1992 to show that the shallower WVZ has the lowest Na_{8.0}, the deeper SMZ has the highest, with the EVZ falling roughly in between (please see revised Fig. 7). We have completely revised and extended the discussion about the relation between spreading rate/style of crustal accretion and geochemical variations along the ridge that we hope the reviewer finds addresses her/his concerns and recommendations.

L110: what I'm personally seeing is a continuous type of magmatism without true time clusters.

There are really too few data to define magmatic cycles as per Fig. 3a. Probability density diagram like that are dependent on the choice of the bins which can influence how many peak one can see. Therefore, looking the individual ages (yellow dots of Fig. 3a), I would rather argue for a very continuous type of magmatism. But it's not about what the author think they visually see, or what I think I visually see – it's about true numbers. What they author would need to do, is to estimate the probability that their “grouping” is not due to chance associated with sample bias.

For example, this graph was created using the RAND function on Excel (generating random number between 0 and 3.8) using 26 data points. Those show artificial group relatively similar to Fig. 3a (and b).

So while I'm NOT saying that there can't be any groups in what the authors shown, such a validating those grouping would require a bit more of a demonstration, preferably using some sort of statistical likelihood calculations, or perhaps correlation with mag anomalies, if any.

I personally think that what we are seeing is not groups, but rather the gaps (i.e., absence of data) between an otherwise continuous dataset due to sample bias. Right or wrong, this has to be better demonstrated. This of course, is not helped by the fact that those are dredge rocks.

Yes, this is clearly a possibility that we cannot rule out. Moreover, sampling was skewed towards sampling younger samples closer to the rift axis. Clearly the $^{40}\text{Ar}/^{39}\text{Ar}$ is too small for robust statistical analysis. We now make it clear that while we see indications of an overall cyclicity we cannot demonstrate this via statistics. So, we have omitted panel A from the original Figure 3 and no longer make any mention of specific cyclical patterns just that the ages are consistent with evidence in the literature for cyclicity. Volcanism must be cyclical in nature. If it was continuous it would not be possible to measure spreading rate based on the rifting of volcanoes and ridges erupting in the rift valley (see methods section for specific examples).

L125 – period of 500 ka. Notwithstanding my remark above, I wonder how this value is calculated? It is really not apparent from Fig. 3 (nor there is a lot of support for such a value). The authors need to give more details on their calculations.

This is no longer relevant now that panel B has been removed. But for the record, we did provide a reference describing the method and a link to the freely available online software we used.

Please note that this comment and response has been moved from later in the review:

L165 – if the mantle melts at 150 km, then one should be able to see garnet bearing mantle signature, which is notoriously visible in the fractionation of REE ratios. Since it is part of the conclusion, such demonstration must be established as well. This is especially true in the case of low degree of partial melting of the MORB mantle since its REE composition will not mask the REE composition from a deeper melt source. That being said, the correlation between the thermal anomaly at 150 km and the spreading rate (Fig. 4a), is much more convincing than the

geochemical correlation.

We do not see an unambiguous signature for a garnet bearing mantle signature. Moreover, locally recorded microearthquakes show that the thickest lithosphere under the SMZ is ≤ 40 km (with thin or absent oceanic crust) below amagmatic sections. So, we are not arguing that the mantle melts at 200 km or 150 km but probably starts at 40 km or shallower based on estimates of lithospheric thickness.

Note that I don't really understand Fig. 4b though, I can't seem to really pinpoint where the Gakkel ridge is on those two velocity graphs.

We now pinpoint where the Gakkel Ridge is on the two velocity graphs by drawing lines between panels (a) and (b).

I found the discussion hard to follow, with a succession of seemingly poorly connected arguments (they are not unrelated at all of course, but it is hard to grasp without some efforts as the transitions are abrupt). This is in contrast to the abstract and conclusions which are much more clear on what is being demonstrated and tested. I'm sure it all makes sense to the authors, of course, but when it is the first time that one read this paper, parts will appear confusing.

For example, the part about the deeper source (Cf. remark above). What are the implications of that? as it stands, it's just an observation, but deep mantle contribution to ridge is potentially a very important observation – that should be developed better to give the reader a sense of why this is important. I suggest rewriting part of the discussion to make it flow better and perhaps keeping it a tad more simple.

Agreed. While keeping it a tad simpler we have developed better the discussion about observation of a deep asthenosphere contribution to the and give ridge the reader a sense of this potentially very important observation.

Lots of the conclusions come from geochemical and geophysical data that have been published and are combined with the geochronology part. It is not like geochron really enlighten some of those data (the slow and fast rates were known before “ ~ 13 mm/yr in the west to only ~ 6 mm/yr in the east”). So, my question is did we learn of the much lower rate in the SMZ in this study, or was that known before as well? If new, I recommend the authors to sell this point better.

The thinking was that SMZ spreading is not different from the overall pattern of steadily decreasing west to east spreading along the ridge. We now follow the helpful suggestion to sell this point better.

Regardless, these data are important because they provide a much more accurate values for the two different spreading rates between WVZ and EVZ vs. SMZ and GRD which will be used by readers in the future. To me the more important conclusion is spreading rate \approx thermal/compositional anomaly of a deeper mantle source.

Agreed. We now try to make and sell this point a lot better.

Reviewer #2 (Remarks to the Author):

The manuscript by O'Connor et al. presents an impressive dataset of $^{40}\text{Ar}/^{39}\text{Ar}$ dates from the Gakkel Ridge. The authors apply this dataset to determine spreading rates along the ridge, then compare the spreading rates to previously documented chemical variations and larger-scale mantle velocity anomalies.

This is an interesting and thought-provoking dataset and should definitely be published in some form (although not necessarily in Nature Communications). I have some concerns and questions regarding the current manuscript, which I outline below:

1. I wonder whether there truly are differences in spreading rate at the WVZ-EVZ versus SMZ-GRD, or whether they simply reflect different styles of crustal accretion. The key to this question appears to be the data from the SMZ. The authors report 10 new dates from this segment. They discard half of these datapoints, which yield robust younger dates, but that the authors argue "do not yield realistic spreading rates due to their mode of emplacement". The authors argue the younger dates reflect eruption of lavas off axis due to magma transport along faults. This seems to be a reasonable interpretation, although I note that it is not supported by evidence beyond the young dates for these lavas, given that these are dredge samples and the context of the samples is not known.

Excluding the younger dates, a linear fit of the SMZ data suggest a slightly slower spreading rate (7.6 mm/yr) than the WVZ and EVZ segments (11.1 mm/yr). I wonder whether this lower rate is robust or whether it instead reflects a wider zone of magmatic accretion along the SMZ. The excluded younger dates suggest there is significant off-axis magmatism (half the dated samples).

We agree. So, we now argue that while the SMZ ages might reflect a slowdown in spreading they might equally well reflect a different style of crustal accretion, e.g., a wide a wider (unfocussed) zone of (cooler) magmatic accretion along the SMZ. Maybe something along the lines outlined in the figure to the left from Langmuir & Forsyth (2007).

Figure 3. Map of the Arctic Ocean's Gakkel Ridge, which is the slowest major spreading ridge on Earth. Spreading rate decreases progressively towards Siberia, as evident from the narrowing of the basin created by the spreading (delimited by the red lines). As spreading rate declines, slower upwelling prevents melting all the way to the surface, and the melting regime becomes progressively truncated, leading to a melting trapezoid rather than a melting triangle such as seen in Figure 1.

The spreading rates calculated using the younger dates from the SMZ should be included on the inset in the lower right of Figure 1. The SMZ data would then form a cloud of data at both slower and faster spreading rates than the WVZ and EVZ.

We agree, especially in the context of our revised model.

Whether or not there are differences in spreading rate between the WVZ-EVZ and SMZ-GRD is important, as much of the manuscript focuses on correlating the variations in spreading rate to other parameters (e.g., chemistry, mantle seismic velocity).

2. The authors argue that the dates record cyclic magmatic trends (Figure 3). This is highly speculative, is not supported by the current data and I would argue this section should be cut from the manuscript. The authors base their argument of a plot of frequency versus date. On this plot, they note that in some cases they have a single date from a given age range (e. g. 0.1 Ma), while at other times they have two samples that give the same date (e.g., 0.3 Ma)—three samples happen to yield a date of ~1.7 Ma. The authors attribute the frequency of a given date to reflect the volume of magmatic activity, where periods with two dated samples reflect

increased magmatism compared to periods with zero or one dated sample.

The number of dated samples from a large geographic area is simply too low to assess temporal variations in the volume of erupted magmas. The apparent "peaks" of magmatic activity may reflect increased magmatic activity, but they may also just reflect random sampling of a uniform age distribution; I consider the latter to be more likely in this case (i.e., the authors happened to date two rocks with an age of 0.3 Ma and one rock with an age of 0.5 Ma). To quantitatively assess temporal variations in the volume of magmatism, the authors would need to date a much larger number of samples, and ideally systematically sample a set geographic area.

Related to this, I note that dates from individual "magmatic cycles" that the authors define and plot in figure 1b, do not all actually overlap within uncertainty (e.g., only two of the three dates in cycle 2 overlap within uncertainty). Other "magmatic cycles" are defined by multiple dates with high uncertainties, where the overlap may reflect the larger analytical uncertainties, rather than a pulse in magmatism (e.g., cycle 5).

As mentioned in our response to Reviewer 1, we agree that the $^{40}\text{Ar}/^{39}\text{Ar}$ dataset is too small for robust statistical analysis. We have omitted panel A from the original Figure 3 and removed any mention of specific cyclical patterns. However, volcanism must be cyclical in nature. If it was continuous it would not be possible to measure spreading rate with so few samples. We now mention that our ages are consistent with evidence in the literature for cyclicity.

3. The authors argue that the distinct spreading rates and chemistry of the WVZ, EVZ, SMZ and GRD may be related to previously observed low shear wave velocity zones, which have been interpreted to reflect elevated asthenospheric temperatures. They suggest that low velocity (higher T) zones below the EVZ and WVZ lead to higher degrees of melting, more robust magmatism and faster spreading. This does not appear to be support by the distribution of the low velocity anomalies. Notably, the low velocity anomalies at 150 km underlies the EVZ and WVZ, but also extend below the dated sample from the GRD and several samples from the SMZ that are adjacent to the EVZ. The GRD and SMZ samples are interpreted to record lower spreading rates, but there is no offset in spreading rate between SMZ samples underlain by the low velocity zone and the other SMZ samples, as would be expected if the presence of the low velocity zone leads to higher spreading rates.

The apparent agreement between the low velocity anomaly at 200 km and the WVZ, SMZ and EVZ boundaries is remarkable. Further, the boundary of the low velocity zone at 200 km coincides with the inferred SMZ – EVZ boundary inferred from differences in spreading rate/style of crustal accretion, which is about 5°E east of initial estimates (34°E) (Michael et al., 2003). Published tomography model resolution is limited by the uneven distribution of seismic stations and constrain less well shallow structures in the mantle. Therefore local surveys have been conducted such as those cited in our paper (Schlindwein et al., 2013; Schlindwein and Schmidt, 2016). We cite the currently most relevant tomography model for the Arctic/Gakkal Ridge by Lebedev et al., (2017) This model uses the latest modelling technology and represents the best information we have at the present. While the reviewer's comment that only the low velocity anomaly at 200 km matches well the WVZ, SMZ and EVZ boundaries is fair, we have a different view/interpretation on the Lebedev et al., tomography data set and argue for a correlation. A definitive test of such a correlation is not possible at present and the seismic stations needed to better resolve the critical mantle area are waiting to be deployed.

We are not arguing that the higher shear wave velocities in the mantle produce faster spreading rates. Our data show constant spreading rates. It is likely that the warmer temperatures are not sufficient to increase the spreading velocities. Moreover, the higher temperatures might provide an explanation for the increased and focussed volcanism along the traverse ridges in the EVZ and the, in general, different topography in the WVZ.

A few quotations from Lebedev et al (2014) follow:

Page 8 below Figure 6

"The solutions of the tomographic inverse problem are non-unique and many different models can fit the data equally well. Specifically, our data tightly constrain the values of V_s in relatively broad depth ranges, but poorly constrain small-scale radial variations in V_s ."

The wavelength of the data is around 230 km ($V_s = 4.62$ km/s, $f = 50$ s). Thus, the tomography can only resolve structures of half of the wavelength (~120 km). In surface tomography of the referenced study the details of (deeper) structures are generally better constrained. This is especially true in the central Arctic, where seismic stations are only located at the rim of the ocean basin. This limits even further the resolution of the shallower parts by the applied algorithms. This provides an explanation for why the low velocity anomalies at 200 km match the WVZ, SMZ and EVZ boundaries better than at 150 km and shallower depths.

Page 11 below Figure 9

"We now highlight a few selected, robust observations, with an emphasis on the more unexpected – or less well-understood – features."

This suggests that Lebedev et al. are very confident that these structures in the mantle exist. Normally tomographers are very reluctant to interpret their results so specifically. Thus, we rely on their results in our paper. We have emphasized this fact more in the revised text in the paragraph "Recent state-of-art tomographic results for the Arctic Ocean offer a robust explanation for the observations along the Gakkel Ridge..."

Page 12 chapter "Eurasia Basin"

"...the ultra-slow-spreading Gakkel Ridge in the Eurasia Basin (e.g. Coakley & Cochran 1998) shows partial melting beneath only a few of its segments..."

AND

"It suggests a major effect of the inflow on the character of the seafloor spreading. The northern limit of the observed low-velocity anomaly (Fig. 4, 80–110 km) is at around 84N and this is where a boundary is observed between the Western Volcanic Zone of the Gakkel Ridge to the south, with basalts covering the seafloor and with well-developed magmatic characteristics similar to those of slow-spreading ridges elsewhere, and the Sparsely Magmatic Zone to the north, with little evidence for magmatism and a predominantly peridotitic crust"

In conclusion, we are adapting interpretations of the most recent tomographic model and relate them to our results in order to provide an explanation for our observations.

The correlation between spreading rate and chemistry appears to be more robust (although I have questions about whether there actually are differences in spreading rates between the segments, as discussed above).

In summary, we now argue that although SMZ spreading might indeed seem be slower based on isotopic dating this finding might be an artifact of a wider, more diffuse (truncated) zone of magmatic accretion that transfers melts to the surface largely via faults.

4. One of the main conclusions of the paper is that "that (slight) differences in mantle temperature/magma supply causes certain segments to spread with slightly different velocities." The authors do not discuss why this might be or what physical process would lead to different spreading rates at segments with higher magma supply?

If spreading rate is controlled by large scale plate movements, it is not clear to me how or why variations in magma supply would impact the spreading rate. This does not seem sustainable over longer timescales. How can the EVZ and WVZ spread at a higher rate than the SMZ? If the plates are spreading at 11.1 mm/yr, how can the SMZ spread at 7 mm/yr over long timescales?

Good point. Please see our expanded and more nuanced discussion in response to a comment above by Reviewer 2. We also make it clearer that different spreading rates can only be a young or unstable process. Otherwise, we should observe large offsets between the segments as found for example along parts of the MAR or EPR.

5. The main contribution of this manuscript is an impressive new $^{40}\text{Ar}/^{39}\text{Ar}$ dataset from the Gakkel Ridge. The authors briefly describe the dating method and do include complete data reports for all the dated samples in the Supplementary material, but there is no real discussion in the main or supplemental text of the $^{40}\text{Ar}/^{39}\text{Ar}$ results. Given that this is the heart of the paper, the authors should include a discussion of the quality of the results, systematics of the $^{40}\text{Ar}/^{39}\text{Ar}$ spectra, etc. Related to this, it is surprising to me that Anthony Koppers is the last author on the paper, when the main contribution of the manuscript is the new $^{40}\text{Ar}/^{39}\text{Ar}$ dates, which were collected in his lab. Given the importance of the $^{40}\text{Ar}/^{39}\text{Ar}$ dates in this manuscript, it should be reviewed by a 40-39 specialist (which I am not).

Please see comments by Reviewer 1, clearly a 40-39 specialist.

Overall, the new contribution of this manuscript is the large 40Ar-39Ar dataset. The authors argue that the dataset suggests there are differences in spreading rate between the EVZ-WVZ and SMZ-GRD. As I outline above, I was not fully convinced by this interpretation and the authors do not discuss a physical basis for why the SMZ-GRD have a lower spreading rate and how that is accommodated with respect to large-scale plate motions.

Please see our revised arguments that should address the reviewer's concerns.

Some of the main conclusion of the manuscript seem disconnected from the new data presented in the paper (i.e., the $^{40}\text{Ar}/^{39}\text{Ar}$ dates). For example, at the end of the abstract, the authors state: "We conclude that relatively slight changes in mantle temperature and composition (volatiles) are likely to cause large variations in the amounts of partial melting and the vigour of boundary layer instabilities, which in turn determines the volume of melt reaching the surface through the Arctic oceanic lithosphere."

We conclude that magmatic and amagmatic seafloor generation in ultra-slow spreading centers reflects the distribution of thermochemical anomalies in the upper mantle."

These are both interesting conclusions, but are based on previously published geochemical and seismic data, rather the results of this study.

Prior to our study magnetic data indicated a steady eastward decrease in spreading rate from 13 mm/yr to 6 mm/yr. Our age data show that spreading is steady in the EVZ and WVZ above different low velocity zones. The few SMZ ages show that there is a change in crustal accretion (maybe spreading) in between. This seems like an interesting finding that is based on $^{40}\text{Ar}/^{39}\text{Ar}$ dates. This correlation is not possible assuming steadily decreasing spreading along the ridge inferred from aeromagnetic measurements and predicted by global models.

If there are changes in spreading rate between the different ridge sections, they may be related to the geochemical and seismic observations; however, the authors do not discuss how these factors could explain or lead to variable spreading rates, beyond noting that the changes in spreading rate appear to correlate with the geochemical and (to a lesser extent) seismic signals. We have expanded our discussion based on different style of magmatism in the SMZ and possibly GRD.

Detailed comments:

Lines 16 to 21: These two sentences both start with "We conclude". Reword?

Done

Lines 42 to 44: "Furthermore, crustal thickness does not become thinner as spreading rate decreases along Gakkel Ridge from ~13 mm/yr in the west to only ~6 mm/yr in the east at 120E Karasik, 1968; Vogt et al. 1979; Jokat and Schmidt-Aursch 2007 and references therein; Schmidt-Aursch and Jokat, 2016."

The data presented in this manuscript show that spreading rate does not decrease along the ridge. It is confusing to present this as fact, when it is actually disproven by the results of the paper.

We have removed this sentence.

Line 66 and throughout: Many in the geochronologists distinguish between date and age. A date is a measured value while an age is the true timing of a geologic event (e.g., The 75 Ma date is interpreted as the age of pluton crystallization). The authors might consider adopting this convention throughout the manuscript.

Done

Lines 85 to 87: "Here we like to consider whether there is a causal relation between spreading rates calculated with the new isotopic ages and the various unexpected findings arising from the AMORE expedition discussed already."

Consider removing "we like" and replacing "unexpected" with unexpected.

Done

Lines 89–91: How are the proposed strike slip faults different from transform faults?

We have removed this statement.

Figure 1: It would be useful to label the dotted line in the spreading rate panel (second from top; e.g., "NNR-MORVEL56 global model")

Done

The excluded younger dates from SMZ should also be included on the Age vs. Profile plot in the lower right.

Done

Figure 1 caption, line 364: Typo., replace "expacement" with emplacement.

Done

Figure 3b. It took me a bit to understand what this plot represents. It is well described in the caption, but the authors should add a label to the y-axis. What criteria was used to define the "magmatic episodes?"

Done

REVIEWER COMMENTS

Reviewer #1 (Remarks to the Author):

Re-Review of "Thermochemical anomalies in the upper mantle control Gakkel Ridge accretion"

Review by Prof. F. Jourdan

The authors went their way to address my initial concerns and I thank them for that. In my view, the paper has been improved as a results and is almost ready for publication. I would not be true to myself if I did not comments on a few points where I still disagree with the authors and they are important points for me, although they do not affect any conclusions in this paper. They are more about what I would call good practice in the field. I hope they will see that bits as a healthy discussion which should not detract of the importance of this paper. All in all, that moved the paper to minor correction needed.

I will only comments on those points that require further discussion here:

ME: Those 4 error ages are not very robust so the reader should be informed within the main text for which of those data this is the case. I would recommend using semi-transparent symbols, to distinguish those from true plateau ages.

AUTHOR RESPONSE: "We have added the following to the methods section:

'A plateau date of 1.65 ± 0.09 Ma for PS59-226-23 is based on 45% of the released ^{39}Ar . However, the inverse isochron age is within analytical error at 2.44 ± 0.58 Ma and the total fusion date of 1.59 ± 0.05 Ma agrees. Five other samples have P-values less than 5% however they all have concordant isochrons and total fusion ages. One of these samples is attributed to fault-related volcanism in the SMZ whereas the other four are equally divided between the WVZ and EVZ. None are used in calculating the average spreading rate in the SMZ. Full analytical results are available in Supplementary Dataset 2."

And the following to the main text:

'The criteria for assessing the quality of the age data are discussed in the methods section. Although six samples do not meet all criteria (Table 1), they still provide robust estimates of the eruption ages as each of them shows concordant plateau, isochron, and total fusion ages; however, we have not used these six lesser-quality ages in our calculations of the slower average spreading rate in the SMZ that we are discussing in the following section, because their inclusion/exclusion doesn't change the outcome of our analyses.'" While using semi-transparent symbols is a good idea we think that this would confuse the non-expert reader. But of course, it's not an issue to make the symbols transparent as might be deemed necessary.

MY RESPONSE AND NEW REQUEST: "A plateau date of ... based on 45% of ^{39}Ar ". But that is the issue here. This is NOT a plateau, this is just a short segment on the age spectrum. I have seen kilotons of those that were not even close to the known age of the sample. So there is no robustness at all in those number, and the fact that they agree with the inverse isochron is not an argument as, provided that the intercept ratio is atmospheric, they should agree with each other regardless of accuracy. Total fusion "age" are meaningless as well since they are just a sum of of all the steps, even if those are discordant. Close to the true age? Possibly, robust? Certainly not. So I am fully opposed to have those error number propagated in the literature as age. I am not sure why the authors do that in fact because as they say themselves, they don't use these data in their calculation. So why not using this opportunity for the authors to show that they are intransigent in their approach to select only the best quality age data. It would reflect well on the paper and increase reader trust in my opinion.

Commen #2

ME: I personally think that what we are seeing is not groups, but rather the gaps (i.e., absence of data) between an otherwise continuous dataset due to sample bias. Right or wrong, this has to be better demonstrated. This of course, is not helped by the fact that those are dredge rocks.

AUTHOR RESPONSE: Yes, this is clearly a possibility that we cannot rule out. Moreover, sampling was skewed towards sampling younger samples closer to the rift axis. Clearly the $^{40}\text{Ar}/^{39}\text{Ar}$ is too small for robust statistical analysis. We now make it clear that while we see indications of an overall cyclicity we cannot demonstrate this via statistics. So, we have omitted panel A from the original Figure 3 and no longer make any mention of specific cyclical patterns just that the ages are consistent with evidence in the literature for cyclicity.

Volcanism must be cyclical in nature. If it was continuous it would not be possible to measure spreading rate based on the rifting of volcanoes and ridges erupting in the rift valley (see methods section for specific examples).

MY RESPONSE: I think we mean different thing by continuous here. I mean continuous on geological time scale. "No cyclicity" in the context of what I mean was that there are no clear eruption sequence that can be identified, and I'm rather pointing toward the fact that the lack of age availability (sample bias) did not mean that there was no activity at this time. Continuous for me does not mean 24/7. It means geologically continuous, but invariably, there are going to be period of on and on volcanism. What I'm doubting here is that this particular dataset can pick that up. In any case, the authors have addressed my concerns.

Reviewer #2 (Remarks to the Author):

This is my second review of the manuscript by O'Connor et al. In my previous review, I highlighted that the manuscript reports an impressive new dataset of $^{40}\text{Ar}/^{39}\text{Ar}$ dates from the Gakkel Ridge, which is a valuable contribution; however, I also raised some concerns regarding the authors conclusions. Here I provide feedback on the revised manuscript.

Overall, the authors clearly put a lot of time and effort into revising the manuscript. I think it is significantly improved, although I am still concerned about some of the issues I raised in my previous review. I would recommend moderate revisions prior to publication.

My main concern with the revised text focuses on whether or not there is a difference in spreading rate between the studied segments. In my previous review, I wrote:

" I wonder whether there truly are differences in spreading rate at the WVZ-EVZ versus SMZ-GRD, or whether they simply reflect different styles of crustal accretion. The key to this question appears to be the data from the SMZ. The authors report 10 new dates from this segment. They discard half of these datapoints, which yield robust younger dates, but that the authors argue "do not yield realistic spreading rates due to their mode of emplacement". The authors argue the younger dates reflect eruption of lavas off axis due to magma transport along faults. This seems to be a reasonable interpretation, although I note that it is not supported by evidence beyond the young dates for these lavas, given that these are dredge samples and the context of the samples is not known.

Excluding the younger dates, a linear fit of the SMZ data suggest a slightly slower spreading rate (7.6 mm/yr) than the WVZ and EVZ segments (11.1 mm/yr). I wonder whether this lower rate is robust or whether it instead reflects a wider zone of magmatic accretion along the SMZ. The excluded younger dates suggest there is significant off-axis magmatism (half the dated samples). If magmas are intruded over a wider zone, lavas erupted at the ridge axis will appear to move off axis more slowly, as some spreading is taken up by off-axis intrusions. In other words, it seems possible that the WVZ and EVZ are characterized by

focused magmatism along the ridge axis, whereas along the SMZ magmatism is distributed across a wider area. At the WVZ and EVZ the focused magmatism along the ridge would lead to the expected increase in dates moving away from the ridge axis. In contrast, at the SMZ, the more diffuse zone of magmatism would lead to some younger dates off axis (as observed in the data). The "apparent" spreading rate for samples erupted on axis would also be slower, because it will take longer for lavas erupted on-axis to move off axis if spreading is accommodated over a wider zone. The same could be true for the GRD, which has only a single dated sample."

The authors agreed with my suggestion and added a robust discussion about the SMZ potentially reflecting a wider zone of accretion. In their response to reviews they wrote: "We agree. So, we now argue that while the SMZ ages might reflect a slowdown in spreading they might equally well reflect a different style of crustal accretion, e.g., a wide a wider (unfocussed) zone of (cooler) magmatic accretion along the SMZ."

In the manuscript text, they conclude:

Line 139: "In conclusion, we define spreading in the SMZ as more likely to reflect a different style of lithospheric/crustal accretion, rather than differences in plate separation, for example by imagining a wider, deeper melting regime producing more diffuse fault-driven magmatic accretion e.g., Langmuir & Forysth, 2007, their Figure 3."

However, despite coming to this conclusion, they refer to the SMZ as having a lower spreading rate throughout the manuscript:

Line 5: "Our age data show that magmatic-dominated sections of the Gakkel Ridge spread at a steady rate of $\sim 11.1 \pm 0.9$ mm/yr whereas amagmatic sections spread $\sim 32\%$ more slowly, which we explain by a different style of crustal accretion."

Line 71: "We can minimise this dredging uncertainty by stacking multiple age-distance profiles when calculating average spreading rates for the different volcanic segments of the ridge: 11 ± 0.9 mm/yr ($n = 15$) for the WVZ and EVZ and 7.6 ± 0.5 mm/yr ($n = 7$) for the SMZ and Gakkel Ridge Deep (GRD)-Laptev Sea (LS) Jokat et al., 2019 (Fig. 1). Thus, notwithstanding the various assumptions and sources of uncertainty (Methods Section), the $^{40}\text{Ar}/^{39}\text{Ar}$ dates provide high precision constraints on spreading rates along ultraslow spreading ridges and variability between different spreading segments, if any. In the case of the Gakkel Ridge we are observing consistently faster spreading in the magmatic WVZ and EVZ segments, opposed to a significantly slower spreading in the amagmatic SMZ and GRD segments (Fig. 1)."

Line 145: "More specifically, we are seeking a relationship between the average spreading rates calculated with the new $^{40}\text{Ar}/^{39}\text{Ar}$ age data and the various unexpected findings arising from the AMORE expedition."

Line 174: "When combining with the tomography models, we deduce that the WVZ and EVZ segments exhibit both a steady and faster spreading, have a thin magmatic lithosphere, and are located above hotter regions in the mantle at depths of 150 km and 200 km, respectively. In contrast, slower spreading and/or differently accreting colder, thick amagmatic lithosphere in the SMZ and GRD-LS corresponds to regions of colder mantle."

Line 328: "The most significant original insight provided by these dates is that, together with this latest tomography model for the Arctic Lebedev et al., 2017, they show that average spreading rates do not vary along most of the Gakkel Ridge, but is lower where there are cold regions in the underlying upper mantle."

Overall, I was left unclear on what exactly the authors' interpretation of the data was. As I said before, they added a very good, robust discussion of how the apparent offset in spreading rates may actually reflect different styles of crustal accretion, but then seem to still interpret the data to reflect different spreading rates throughout the manuscript.

I would reiterate that I think the case for the SMZ having a different spreading rate is weak. The authors

calculate the slower spreading rate by discarding half of the dated samples from the SMZ. This is highlighted in the revised figure 1 inset. The light blue and dark blue data points come from the SMZ, with the authors using the dark blue data points to argue for a slower spreading rate. An average of all the data would yield a rate similar to the WVZ and EVZ. As I outline in my previous comments, this is consistent with magmatism across a more diffuse zone.

I would recommend the authors further revise the text to more clearly reflect their preferred model for the SMZ 40-39 dates. If they agree that the distribution of dates reflects a wider zone of accretion or different style of accretion, rather than a slower spreading rate, it would make sense to be consistent throughout the manuscript.

Instead of first presenting the SMZ as having a slower rate and then discussing why that might be, I think it would be clearer just to say the data from the SMZ do not fall along a single spreading rate. The current text still leaves the impression that there are differences in spreading rate along the length of the ridge, but I would argue this is not supported by the 40-39 dates. The SMZ data only suggest a slow rate if half of the dates are excluded.

My other general comment is that most of the manuscript's discussion section is largely disconnected from the new 40-39 dates presented in the paper. The discussion presents a detailed discussion of geophysical and geochemical differences between the different segments of the Gakkel ridge. This is interesting, but fundamentally is not driven by new insight from the 40-39 data. The discussion is largely based on published data and could have been written with or without the 40-39 results. I would say that the 40-39 data align with the already published geophysical and geochemical results, but do not really provide new insight into large-scale mantle dynamics.

In addition to these more general comments, I provide a few more detailed comments below:

Lines 12–14: The authors should add references to support this statement.

Lines 29–33: This section discusses the spreading rate decreasing systematically along the Gakkel ridge. The authors should discuss what this previous interpretation was based on.

Line 94: "For example, the isotopic dataset hints that volcanism at the ridge might be cyclical (Supplementary Figure 1)."

Both myself and the second review strongly disagreed with this interpretation of the data. There is not enough data to provide evidence for cyclic magmatic trends. The authors removed much of the text, but this sentence should be removed as well.

Lines 107–108: What is this based on? There is no reference.

Figures 2–5: Some of the text is too small to read in these figures. I also wonder why the authors include both higher and lower resolution bathymetry? I am not sure the lower resolution data add anything.

Figure 6: The panels on the right are pixelated. The authors should try to obtain higher resolution versions of these figures.

Figure 7: This figure is significantly improved from the previous draft. However, the Figure should include a panel showing Na, as the Na content of the different segments is discussed in detail in the text. In the current figure there is a low-resolution image of an Na plot in the lower left, but I am not sure if this is a mistake or is supposed to be a part of the figure. If it is supposed to be part of the figure, the authors

should replot the data to be consistent with the rest of the plots in the figure.

REVIEWER COMMENTS

We thank both reviewers for further constructive and insightful reviews.

Reviewer #1 (Remarks to the Author):

Re-Review of “Thermochemical anomalies in the upper mantle control Gakkel Ridge accretion”
Review by Prof. F. Jourdan

The authors went their way to address my initial concerns and I thank them for that. In my view, the paper has been improved as a results and is almost ready for publication. I would not be true to myself if I did not comments on a few points where I still disagree with the authors and they are important points for me, although they do not affect any conclusions in this paper. They are more about what I would call good practice in the field. I hope they will see that bits as a healthy discussion which should not detract of the importance of this paper. All in all, that moved the paper to minor correction needed.

We thank Prof. Jourdan for his very helpful comments. We are very pleased that he considers that our manuscript is almost ready for publication.

I will only comments on those points that require further discussion here:

ME: Those 4 error ages are not very robust so the reader should be informed within the main text for which of those data this is the case. I would recommend using semi-transparent symbols, to distinguish those from true plateau ages.

AUTHOR RESPONSE: “We have added the following to the methods section:

“A plateau date of 1.65 ± 0.09 Ma for PS59-226-23 is based on 45% of the released ^{39}Ar . However, the inverse isochron age is almost within analytical error at 2.44 ± 0.58 Ma and the total fusion date of 1.59 ± 0.05 Ma agrees. Five other samples have P-values less than 5% however they all have concordant isochrons and total fusion ages. One of these samples is attributed to fault-related volcanism in the SMZ whereas the other four are equally divided between the WVZ and EVZ. None are used in calculating the average spreading rate in the SMZ. Full analytical results are available in Supplementary Dataset 2.”

And the following to the main text:

“The criteria for assessing the quality of the age data are discussed in the methods section. Although six samples do not meet all criteria (Table 1), they still provide robust estimates of the eruption ages as each of them shows concordant plateau, isochron, and total fusion ages; however, we have not used these six lesser-quality ages in our calculations of the slower average spreading rate in the SMZ that we are discussing in the following section, because their inclusion/exclusion doesn’t change the outcome of our analyses.” While using semi-transparent symbols is a good idea we think that this would confuse the non-expert reader. But of course, it’s not an issue to make the symbols semi-transparent as might be deemed necessary.

MY RESPONSE AND NEW REQUEST:

“A plateau date of ... based on 45% of ^{39}Ar ”. But that is the issue here. This is NOT a plateau, this is just a short segment on the age spectrum. I have seen kilotons of those that were not even close to the known age of the sample. So there is no robustness at all in those number, and the fact that they agree with the inverse isochron is not an argument as, provided that the intercept ratio is atmospheric, they should agree with each other regardless of accuracy.

Total fusion “age” are meaningless as well since they are just a sum of of all the steps, even if those are discordant. Close to the true age? Possibly, robust? Certainly not. So, I am fully opposed to have those error number propagated in the literature as age. I am not sure why the authors do that in fact because as they say themselves, they don’t use these data in their calculation. So why not using this opportunity for the authors to show that they are intransigent in their approach to select only the best quality age data. It would reflect well on the paper and increase reader trust in my opinion.

We now clearly identify the 7 samples that have a plateau with less than 50% ^{39}Ar and/or P-values $> 5\%$ in Table 1 and Figure 1. Using semi-transparent symbols didn’t work out so we use a unique symbol (diamond). The above-quoted paragraphs from the main text have been changed as follows:

Methods section:

“The quality of a $^{40}\text{Ar}/^{39}\text{Ar}$ step-heating experiment is assessed based on the following criteria: an acceptable age plateau (1) includes at least 50% of the gas released, (2) has a mean square weighted deviation (MSWD) of approximately 1.0 and within the statistically allowed upper limit, (3) shows an inverse isochron with a $^{40}\text{Ar}/^{36}\text{Ar}$ intercept of about $295.5 \pm 2\sigma$, and (4) has a concordant plateau, isochron, and total fusion ages and (5) a p-value of $>5\%$. Seven samples have a mini-plateau with less than 50% ^{39}Ar and/or a P-values $> 5\%$ (see Table 1 and Figure 1). Full analytical results are available in Supplementary Dataset 2.”

Main text:

“Seven samples do not meet one or both of the criteria that a plateau consists of at least 50% ^{39}Ar and has a P-value $> 5\%$ (Table 1 and Fig. 1). These seven samples, therefore, cannot be considered robust but rather approximate age estimates.”

Comment #2

ME: I personally think that what we are seeing is not groups, but rather the gaps (i.e., absence of data) between an otherwise continuous dataset due to sample bias. Right or wrong, this has to be better demonstrated. This of course, is not helped by the fact that those are dredge rocks.

AUTHOR RESPONSE: Yes, this is clearly a possibility that we cannot rule out. Moreover, sampling was skewed towards sampling younger samples closer to the rift axis. Clearly the $^{40}\text{Ar}/^{39}\text{Ar}$ is too small for robust statistical analysis We now make it clear that while we see indications of an overall cyclicity we cannot demonstrate this via statistics. So, we have omitted panel A from the original Figure 3 and no longer make any mention of specific cyclical patterns just that the ages are consistent with evidence in the literature for cyclicity. Volcanism must be cyclical in nature. If it was continuous it would not be possible to measure spreading rate based on the rifting of volcanoes and ridges erupting in the rift valley (see methods section for specific examples).

MY RESPONSE: I think we mean different thing by continuous here. I mean continuous on geological time scale. “No cyclicity” in the context of what I mean was that that there are no clear eruption sequence that can be identified, and I’m rather pointing toward the fact that the lack of age availability (sample bias) did not means that there was no activity at this time. Continuous for me does not mean 24/7. It means geologically continuous, but invariably, there are going to be period of on and on volcanism. What I’m doubting here is that this particular dataset can pick that up. In any case, the authors have addressed my concerns.

We agree with Prof. Jordan that our limited dataset can robustly pick up cyclicity. Our point is that our data are consistent with the notion of cyclicity, which is well established by the literature we cite.

Nevertheless, to address the continued concern of both reviewers we have removed the sentence "For example, the isotopic dataset hints that volcanism at the ridge might be cyclical (Supplementary Figure 1)."

Reviewer #2 (Remarks to the Author):

This is my second review of the manuscript by O'Connor et al. In my previous review, I highlighted that the manuscript reports an impressive new dataset of $^{40}\text{Ar}/^{39}\text{Ar}$ dates from the Gakkel Ridge, which is a valuable contribution; however, I also raised some concerns regarding the authors conclusions. Here I provide feedback on the revised manuscript.

Overall, the authors clearly put a lot of time and effort into revising the manuscript. I think it is significantly improved, although I am still concerned about some of the issues I raised in my previous review. I would recommend moderate revisions prior to publication.

We thank Reviewer 2 for very helpful comments and suggestions. We are very pleased that s/he considers that our manuscript is improved significantly.

My main concern with the revised text focuses on whether or not there is a difference in spreading rate between the studied segments. In my previous review, I wrote:

" I wonder whether there truly are differences in spreading rate at the WVZ-EVZ versus SMZ-GRD, or whether they simply reflect different styles of crustal accretion. The key to this question appears to be the data from the SMZ. The authors report 10 new dates from this segment. They discard half of these datapoints, which yield robust younger dates, but that the authors argue "do not yield realistic spreading rates due to their mode of emplacement".

The authors argue the younger dates reflect eruption of lavas off axis due to magma transport along faults. This seems to be a reasonable interpretation, although I note that it is not supported by evidence beyond the young dates for these lavas, given that these are dredge samples and the context of the samples is not known.

Excluding the younger dates, a linear fit of the SMZ data suggest a slightly slower spreading rate (7.6 mm/yr) than the WVZ and EVZ segments (11.1 mm/yr). I wonder whether this lower rate is robust or whether it instead reflects a wider zone of magmatic accretion along the SMZ.

The excluded younger dates suggest there is significant off-axis magmatism (half the dated samples). If magmas are intruded over a wider zone, lavas erupted at the ridge axis will appear to move off axis more slowly, as some spreading is taken up by off-axis intrusions. In other words, it seems possible that the WVZ and EVZ are characterized by focused magmatism along the ridge axis, whereas along the SMZ magmatism is distributed across a wider area.

At the WVZ and EVZ the focused magmatism along the ridge would lead to the expected increase in dates moving away from the ridge axis. In contrast, at the SMZ, the more diffuse zone of magmatism would lead to some younger dates off axis (as observed in the data). The "apparent" spreading rate for samples erupted on axis would also be slower, because it will take longer for lavas erupted on-axis to move off axis if spreading is accommodated over a wider zone. The same could be true for the GRD, which has only a single dated sample."

The authors agreed with my suggestion and added a robust discussion about the SMZ potentially reflecting a wider zone of accretion. In their response to reviews they wrote: "We agree. So, we now

argue that while the SMZ ages might reflect a slowdown in spreading they might equally well reflect a different style of crustal accretion, e.g., a wide a wider (unfocussed) zone of (cooler) magmatic accretion along the SMZ."

In the manuscript text, they conclude:

Line 139: "In conclusion, we define spreading in the SMZ as more likely to reflect a different style of lithospheric/crustal accretion, rather than differences in plate separation, for example by imagining a wider, deeper melting regime producing more diffuse fault-driven magmatic accretion e.g., Langmuir & Forysth, 2007, their Figure 3."

However, despite coming to this conclusion, they refer to the SMZ as having a lower spreading rate throughout the manuscript:

Line 5: "Our age data show that magmatic-dominated sections of the Gakkel Ridge spread at a steady rate of $\sim 11.1 \pm 0.9$ mm/yr whereas amagmatic sections spread $\sim 32\%$ more slowly, which we explain by a different style of crustal accretion."

We have revised the above sentence as follows:

' Our age data show that magmatic-dominated sections of the Gakkel Ridge spread at a steady rate of $\sim 11.1 \pm 0.9$ mm/yr whereas amagmatic sections have a more widely distributed melt supply yielding ambiguous spreading rate information..'

Line 71: "We can minimise this dredging uncertainty by stacking multiple age-distance profiles when calculating average spreading rates for the different volcanic segments of the ridge: 11 ± 0.9 mm/yr ($n = 15$) for the WVZ and EVZ and 7.6 ± 0.5 mm/yr ($n = 7$) for the SMZ and Gakkel Ridge Deep (GRD)-Laptev Sea (LS) Jokat et al., 2019 (Fig. 1). Thus, notwithstanding the various assumptions and sources of uncertainty (Methods Section), the $40\text{Ar}/39\text{Ar}$ dates provide high precision constraints on spreading rates along ultraslow spreading ridges and variability between different spreading segments, if any. In the case of the Gakkel Ridge we are observing consistently faster spreading in the magmatic WVZ and EVZ segments, opposed to a significantly slower spreading in the amagmatic SMZ and GRD segments (Fig. 1)."

We have revised the above paragraph as follows:

'We can minimise this dredging uncertainty by stacking multiple age-distance profiles when calculating average spreading rates for the different segments of the ridge (Fig. 1). We argue that, notwithstanding the various assumptions and sources of uncertainty (Methods Section), the $40\text{Ar}/39\text{Ar}$ dates provide high precision constraint on spreading rates along the magmatic WVZ and EVZ sections of the ridge: 11 ± 0.9 mm/yr ($n = 15$) (Fig. 1). In contrast, the $40\text{Ar}/39\text{Ar}$ dates for the SMZ and Gakkel Ridge Deep (GRD)-Laptev Sea (LS) Jokat et al., 2019 group at faster and slower spreading rates, rather than falling along a single spreading rate (Fig. 1).

The authors calculate the slower spreading rate by discarding half of the dated samples from the SMZ. This is highlighted in the revised figure 1 inset. The light blue and dark blue data points come from the SMZ, with the authors using the dark blue data points to argue for a slower spreading rate. An average of all the data would yield a rate similar to the WVZ and EVZ. As I outline in my previous comments, this is consistent with magmatism across a more diffuse zone.

Overall, I was left unclear on what exactly the authors' interpretation of the data was. As I said before, they added a very good, robust discussion of how the apparent offset in spreading rates may

actually reflect different styles of crustal accretion, but then seem to still interpret the data to reflect different spreading rates throughout the manuscript.

I would reiterate that I think the case for the SMZ having a different spreading rate is weak.

I would recommend the authors further revise the text to more clearly reflect their preferred model for the SMZ 40-39 dates.

We understand and appreciate all the reviewer's points and concerns about the notion that the evidence for slower spreading is not robust. But we think that it is necessary to show this trend to readers while making it clear that it is poorly constrained. We argue that it reflects a different style of accretion rather than plate separation. We are discussing here completely new insights that in our view are extremely important, even if errors might be large. We need to keep in mind the much larger error in magnetic age models or the geodetic kinematic model derived from GPS data that cannot resolve such details. Please see the following paragraph we have added to the main text.

We have added the following paragraph to the main text:

'Four SMZ and single Gakkel Ridge Deep (GRD) samples predict a poorly constrained slower spreading rate: 7.6 ± 0.5 mm/yr ($n = 7$) (Fig. 1). The other SMZ dated samples predicting unrealistically fast spreading rates were dredged from near the base of the rift valley wall (Fig. 3) suggesting there is significant off-axis magmatism (half the dated samples). This age distribution is consistent with a wider zone of accretion rather than a slower spreading rate. The 'apparent' spreading rate for samples erupted on-axis would be slower because it will take longer for lavas erupted on-axis to move off-axis if spreading is accommodated over a wider zone. The same could be true for the GRD, which is based on a single dated sample. However, regressing all the SMZ age data also yields a slower spreading rate (8.1 mm/yr) compared to the WVZ and EVZ so it cannot be excluded that these dates might belong to two different groups. Moreover, none of the dates lies in vicinity of the WVZ/EVZ regression, even when considering the error bars. Thus, while we cannot link the slower SMZ-GRD trend necessarily to plate separation, it may well prove to be significant for understanding amagmatic crustal accretion.'

Line 145: "More specifically, we are seeking a relationship between the average spreading rates calculated with the new $^{40}\text{Ar}/^{39}\text{Ar}$ age data and the various unexpected findings arising from the AMORE expedition."

Changed as follows:

'More specifically, we are seeking a relationship between the new $^{40}\text{Ar}/^{39}\text{Ar}$ age data, (especially the data that are inconsistent with spreading ages predicted by marine magnetic data) and the various unexpected findings arising from the AMORE expedition.'

Line 174: "When combining with the tomography models, we deduce that the WVZ and EVZ segments exhibit both a steady and faster spreading, have a thin magmatic lithosphere, and are located above hotter regions in the mantle at depths of 150 km and 200 km, respectively. In contrast, slower spreading and/or differently accreting colder, thick amagmatic lithosphere in the SMZ and GRD-LS corresponds to regions of colder mantle."

Changed as follows: Line 174:

" Thus, $^{40}\text{Ar}/^{39}\text{Ar}$ dates reveal that spreading rate/accretion style correlates also with lithospheric thickness.'

Line 328: "The most significant original insight provided by these dates is that, together with this latest tomography model for the Arctic Lebedev et al., 2017, they show that average spreading rates do not vary along most of the Gakkel Ridge, but is lower where there are cold regions in the underlying upper mantle."

Changed as follows:

"The most significant original insight provided by these dates is that, together with this latest tomography model for the Arctic Lebedev et al., 2017, they show that steady spreading rate correlates with hotter regions in the underlying upper mantle. Whereas amagmatism reflects colder underlying upper mantle.

My other general comment is that most of the manuscript's discussion section is largely disconnected from the new 40-39 dates presented in the paper.

The discussion presents a detailed discussion of geophysical and geochemical differences between the different segments of the Gakkel ridge. This is interesting, but fundamentally is not driven by new insight from the 40-39 data.

The discussion is largely based on published data and could have been written with or without the 40-39 results. I would say that the 40-39 data align with the already published geophysical and geochemical results, but do not really provide new insight into large-scale mantle dynamics.

The discussion is now much shorter and we link the revised text and the 40-39 data clearer. We make it clearer that we show for the first time that there is a relation between the spreading rate and hotter regions in the mantle. We show also that this relationship breaks down above colder regions. This relationship could not be postulated before our study because of the wide assumption that the spreading rate decreases steadily along the length of the ridge. We are unaware where in the literature high precision 40-39 data show a connection between spreading rate and mantle temperature? Our new ⁴⁰Ar/³⁹Ar data for the Gakkel Ridge shows that spreading rate/accretion style correlates with locations of thermochemical anomalies in the asthenosphere beneath the ridge. Evidence that the structure of the lithosphere, the extent of magmatism, and its composition correlate with spreading rate/style of accretion links them in turn to mantle temperature.

In addition to these more general comments, I provide a few more detailed comments below:

Lines 12–14: The authors should add references to support this statement.

We have deleted this sentence

Lines 29–33: This section discusses the spreading rate decreasing systematically along the Gakkel ridge. The authors should discuss what this previous interpretation was based on.

This is explained further down in the text.

Line 94: "For example, the isotopic dataset hints that volcanism at the ridge might be cyclical (Supplementary Figure 1)."

Both myself and the second review strongly disagreed with this interpretation of the data. There is not enough data to provide evidence for cyclic magmatic trends. The authors removed much of the text, but this sentence should be removed as well.

Done

Lines 107–108: What is this based on? There is no reference.

Done

Figures 2–5: Some of the text is too small to read in these figures. I also wonder why the authors include both higher and lower resolution bathymetry? I am not sure the lower resolution data add anything.

The maps are all made using the same compilation of available bathymetry for the Gakkel Ridge / Arctic Ocean. Coverage is patchy giving the impression of higher and lower resolution. It's a good point about some of the text being too small. We've increased the font size as much as possible without making the figures too busy.

Figure 6: The panels on the right are pixelated. The authors should try to obtain higher resolution versions of these figures.

Done

Figure 7: This figure is significantly improved from the previous draft. However, the Figure should include a panel showing Na, as the Na content of the different segments is discussed in detail in the text. In the current figure there is a low-resolution image of an Na plot in the lower left, but I am not sure if this is a mistake or is supposed to be a part of the figure. If it is supposed to be part of the figure, the authors should replot the data to be consistent with the rest of the plots in the figure.

We agree that the resolution Na figure needs to be increased. We do not have the data available to replot it so we reference it in Michael et al. (2003).